# SpaCE-Eval: A Benchmark for Real-World Multi-Modal Reasoning

**Xuyou Yang**[1], **Yucheng Zhao**[2], **Wenxuan Zhang**[1*], **Immanuel Koh**[1]

[1]Singapore University of Technology and Design
[2]Alibaba Group

```
xuyou_yang@mymail.sutd.edu.sg
yucheng.zhao@alibaba-inc.com
{wxzhang, immanuel_koh}@sutd.edu.sg
```

## Abstract

Multi-modal Large Language Models (MLLMs) represent a significant advancement in artificial intelligence. Among the growing capabilities exhibited by MLLMs, abilities to understand and reason in real-world environments stand out as particularly vital as a fundamental prerequisite for a wide array of real-world applications. The current methods for evaluating MLLMs often fall short in their ability to comprehensively assess these crucial capabilities. However, being able to reason on complex environment-scale spaces, for example, room spaces, building spaces, and even urban spaces, and to predict the future and plan actions, is essential for humans and various autonomous agents to survive in the real physical world. To address these gaps, we propose a visual-question-answering benchmark, SpaCE-Eval (**Spa**tial Reasoning, **C**ommonsense Knowledge and **E**nvironment Interaction), designed to evaluate MLLM's reasoning abilities in real-world environments. As the name suggests, it challenges the models to reason on complex spatial scenarios, invoke commonsense knowledge of the physical world, and interact with the environment. The dataset consists of all new diagrams purposefully produced by humans, where diagram-question pairs are meticulously refined and selected through a rigorous pipeline. Additionally, with the benchmark, we evaluate a selection of leading MLLMs, both proprietary and open source. The results suggest that significant enhancement of MLLMs in reasoning in the real physical world is necessary to realise more advanced general artificial intelligence. Code and dataset available at `https://github.com/xuyou-yang/SpaCE-Eval`.

## 1 Introduction

Multi-modal Large Language Models (MLLMs) have advanced rapidly in recent years, showing growing capabilities in tasks that require joint visual and textual comprehension. Existing models, either commercial models such as GPT-4o (Hurst et al., 2024), Grok (xAI, 2024) and Gemini (Gemini Team Google et al., 2023), or open-source ones such as Qwen2.5-VL (Bai et al., 2025), Llama 4 (Meta AI, 2025) and LLaVA-OneVision (Li et al., 2024a), have achieved impressive performance on diverse visual question answering (VQA) tasks (Goyal et al., 2017; Masry et al., 2022; Singh et al., 2019; Mathew et al., 2021; Lu et al., 2023).

As MLLMs are increasingly deployed in real-world applications, including robotics (Driess et al., 2023; Li et al., 2024b; Yue et al., 2024b), autonomous navigation (Shah et al., 2023; Zhou et al., 2024), and embodied agents (Szot et al., 2024), it becomes critical to assess **whether these models can reason effectively in dynamic physical environments**, which require spatial awareness, commonsense understanding, and interaction with the environment.

A number of benchmarks have been proposed to evaluate MLLMs' visual reasoning abilities, but existing efforts fall short in several important aspects. First, existing spatial understanding and

---

*Corresponding author.

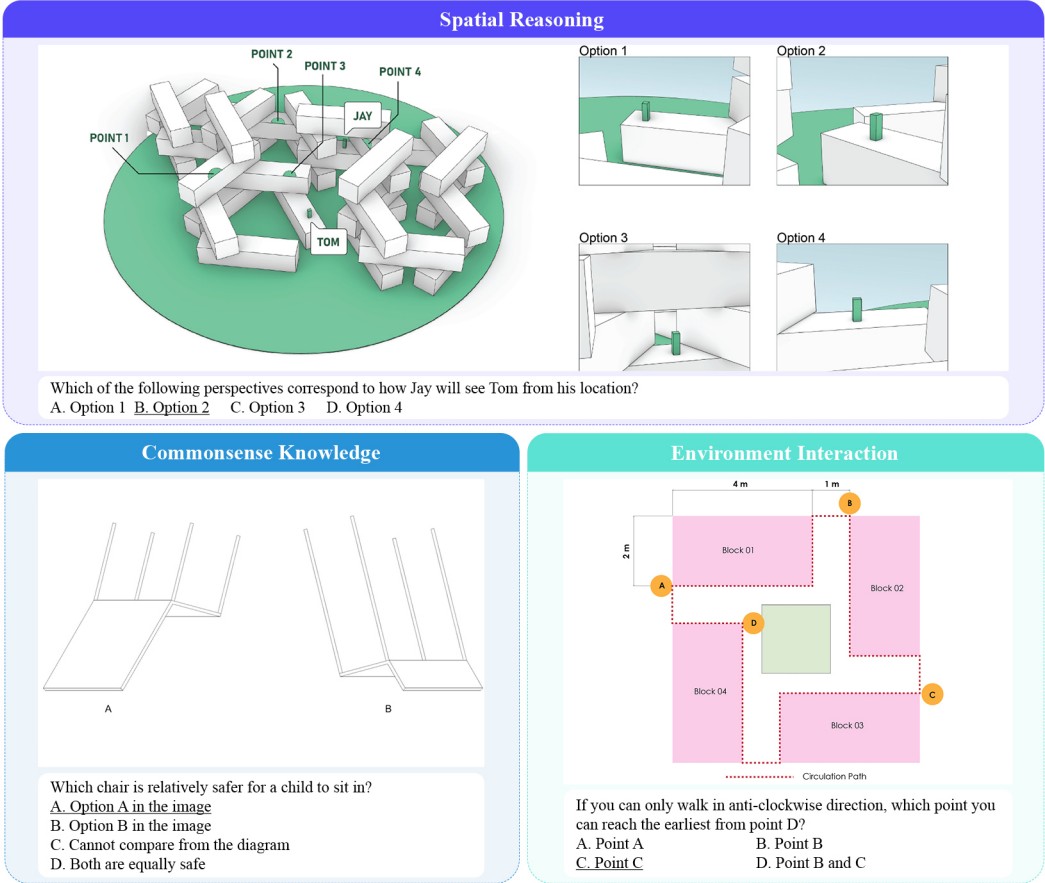

Figure 1: VQA examples of the three categories in SpaCE-Eval. The ground truth is indicated with underline.

reasoning datasets (Wang et al., 2024; Chen et al., 2024; Liu et al., 2023; Cheng et al., 2024) often focus on object scale understanding (e.g., household objects, table games), ignoring that the environment we live in has much more diverse scales, ranging from a room, a building, to a community and even a city. More importantly, the tasks are typically simple, such as object counting, position understanding, and relative relationships (e.g., left or right, close or far etc.), whereas spatial relationships are more complicated in the real world, and require different types of reasoning abilities. Second, while some datasets probe physical or social commonsense, for example, PIQA (Bisk et al., 2020), VisualCOMET (Park et al., 2020) and CulturalVQA (Nayak et al., 2024), they typically lack spatial grounding, making it difficult to evaluate how well MLLMs integrate such knowledge with physical context. Third, most existing datasets assess a static understanding of a given image, instead of reasoning about options or actions to interact with the environment. However, predicting and planning for what will happen next is essential for humans and autonomous agents to interact with complex environments to survive.

To address these limitations, we introduce **SpaCE-Eval** (**Spa**tial Reasoning, **C**ommonsense Knowledge and **E**nvironment Interaction), a new benchmark designed to evaluate MLLM's capability to reason in real-world environments. As shown in Figure 1, SpaCE-Eval consists of three categories: (1) **Spatial Reasoning** assesses models' spatial reasoning abilities in environments on multiple scales. It requires the MLLMs to comprehensively reason on spaces which have complex spatial configurations and relationships in real-world scenarios.; (2) **Commonsense Knowledge** tests the MLLM's background knowledge necessary to conduct reasoning in the real-world spaces at the commonsense level. (2) **Environment Interaction** evaluates MLLMs' ability to compare options, make decisions and predict affordances in real-world conditions, as a user or a decision maker, in order to better interact with the environment. The scale of spaces and reasoning subjects utilised in all categories ranges from small items, rooms, buildings, to urban contexts. Together, these three categories target core competencies required to deploy MLLMs in a real physical world.

To construct SpaCE-Eval, we employ a rigorous pipeline. First, human experts with design backgrounds draw brand-new diagrams for every task category. Different from statistical charts, these new diagrams are info-graphics. This approach offers two key benefits: (i) all diagrams are freshly created, eliminating data-contamination risks and preventing models from relying on prior exposure to publicly available diagrams; and (ii) by encouraging contributors to follow professional standards while retaining their personal drawing styles, we achieve greater visual diversity. Next, two carefully crafted questions are written for every diagram, each with text or visual answer choices. Each item then undergoes a three-stage quality process: peer review for clarity and correctness, adversarial rewriting to remove linguistic shortcuts, and multiple rounds of verification by expert meta-annotators. The final release contains 1,139 high-quality image-question pairs across the benchmark's three main categories.

Based on the newly introduced SpaCE-Eval benchmark, we evaluate a wide range of proprietary and open-source MLLMs, covering different model families and sizes. SpaCE-Eval appears to be very challenging across all sorts of models, especially with the Spatial Reasoning category. For instance, the best overall result is only 56.37% across all tested models, while the highest accuracy in Spatial Reasoning is merely 42.25% both achieved by GPT-5 (OpenAI, 2025). We also observe that models perform significantly worse as the spatial scale increases, and visual understanding consistently lags behind textual comprehension. Moreover, analysis reveals that many models rely heavily on surface-level patterns rather than engaging with deeper conceptual or spatial structures. These findings underscore the need for improved multi-modal reasoning capabilities and highlight SpaCE-Eval as a robust benchmark for measuring progress in real-world MLLM reasoning.

## 2 RELATED WORK

**MLLMs** The remarkable progress of large language models (LLMs) has driven widespread adoption of the Transformer architecture (Vaswani et al., 2017) in the computer vision domain, leading to the development of models like ViT (Dosovitskiy et al., 2020), CLIP (Radford et al., 2021), and MAE (He et al., 2022). Leveraging the foundational capabilities of LLMs, MLLMs such as GPT-4o (Hurst et al., 2024), Qwen2.5-VL (Bai et al., 2025), LLaVa (Liu et al., 2023) integrate information across multiple modalities and have shown strong generalisation across a wide array of tasks. In particular, these models have demonstrated increasingly sophisticated reasoning capabilities in real-world environments, where understanding spatial relationships and physical constraints is critical. In addition to excelling at tasks like different sorts of VQA (Yue et al., 2024a; Singh et al., 2019; Mathew et al., 2021; Masry et al., 2022) and mathematical reasoning (Lu et al., 2023), these models show growing competence in spatial (Cheng et al., 2024; Chen et al., 2024) and commonsense (Park et al., 2020; Nayak et al., 2024) reasoning, marking a significant step toward grounded intelligence in complex physical contexts.

**Benchmarks** Our work is mostly related to benchmarks and datasets that cover one or more aspects of the three categories in SpaCE-Eval. We compare some representative benchmarks related to our work in Table 1. Firstly, existing benchmarks on spatial reasoning (Wang et al., 2024; Chen et al., 2024; Liu et al., 2023; Cheng et al., 2024) focus on relative spatial relationships between objects, such

Table 1: Comparison of representative benchmarks in three categories.

| Benchmark | Spatial Reasoning | Commonsense Knowledge | Environment Interaction |
|---|---|---|---|
| Spatial VQA (Chen et al., 2024) | Simple relative location, distance, height, etc. | – | – |
| SpatialEval (Wang et al., 2024) | Spatial map, maze, grid, etc. | – | – |
| GRASP (Jassim et al., 2023) | – | Object grounding and intuitive physics | – |
| CulturalVQA (Nayak et al., 2024) | – | Geo-diverse cultural understanding | – |
| VisualCOMET (Park et al., 2020) | – | Visual commonsense | Events before and after, human intent |
| CLEVRER (Yi et al., 2019) | Object trajectories, relative positions | Physical commonsense (e.g., collisions, feasibility) | – |
| **Ours** | **Complex spatial relations of multiple spatial scales** | **Intuitive science, engineering, physics, and culture** | **Interaction with environment** |

as relative position, adjacency and orientation. Benchmarks like SptialEval (Wang et al., 2024) test models' ability to perform in table games, for example, navigate in mazes. Despite the challenging spatial reasoning process, these are not the real physical world. Secondly, benchmarks (Jassim et al., 2023; Meng et al., 2024) that assess the commonsense knowledge of the models mainly concentrate on understanding physical features and laws, and do not contain cultural context (Nayak et al., 2024) that is also an important aspect of the real world. Lastly, datasets (Shridhar et al., 2020; Padmakumar et al., 2022; Fan et al., 2022) on interaction in various environments are mostly used to evaluate or enhance embodied agents given instructions. In addition to these single-category benchmarks, some benchmarks (Park et al., 2020; Yi et al., 2019) cover two areas of the three.

## 3 THE SPACE-EVAL BENCHMARK

We introduce the SpaCE-Eval, a novel and meticulously curated benchmark to assess the reasoning abilities of MLLMs in real physical world environments. The benchmark challenges the abilities of MLLMs to perform complex reasoning on multiple scales of spaces. It provides questions which require the integration of abstract visual perception, spatial simulation, commonsense knowledge and deliberate reasoning abilities to answer. These abilities are essential for humans and autonomous agents to navigate and survive in real physical world.

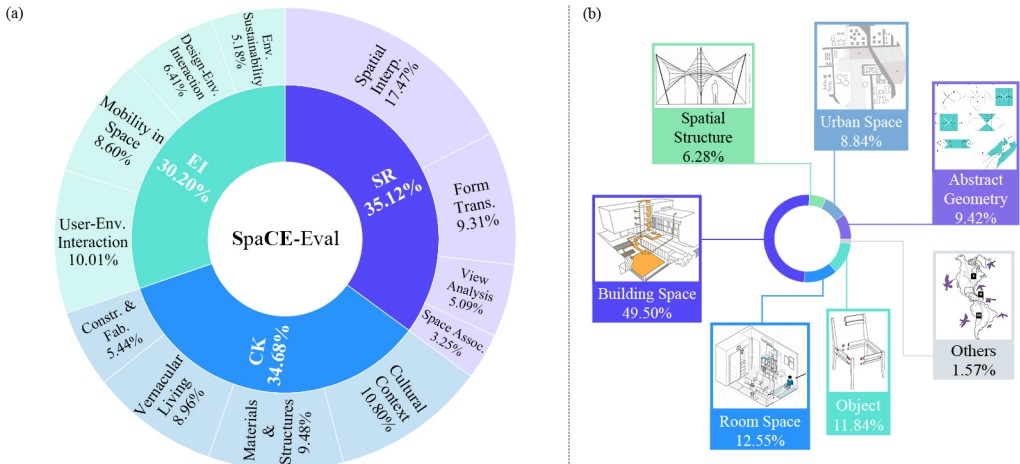

Figure 2: Overview of the dataset. (a) The dataset comprises three main categories: Spatial Reasoning (SR), Commonsense Knowledge (CK), Environment Interaction (EI), and twelve subcategories, with each category assigned a corresponding data ratio. (b) The benchmark challenges models to reason on a wild range of spatial scales.

### 3.1 DATASET DESIGN

Considering the abilities and background knowledge required to understand and reason in the real world, we design three parallel main categories of the dataset: Spatial Reasoning, Commonsense Knowledge and Environment Interaction. To obtain more detailed insights, there are also four subcategories within each main category. Figure 2 illustrates the composition of the dataset. Appendix A.0.1 displays examples of each category and subcategory.

**Spatial Reasoning** This category assesses models' fundamental spatial reasoning abilities in real world spaces with complex spatial relationships and of various scales, from rooms, buildings, to urban contexts. It requires the MLLMs to comprehensively interpret spaces through various complicated spatial reasoning process, including (i) Spatial Interpretation: reasoning one correct perspective/view from a given angle or viewpoint, including indoor and outdoor, static and dynamic conditions; (ii) Space Association: associating spaces by linking different views (plan, section, or elevation); (iii) View Analysis: analysing visibility of items or spaces in complex spatial setups; and (iv) Form Transformation: predicting the new form under explicitly or implicitly given transformation rules.

**Commonsense Knowledge**   This category tests the MLLMs' background knowledge associated with spaces necessary to conduct reasoning in the real world at the commonsense or intuitive level. It consists of a wide range of fields categorised into four groups: (i) Material and Science: assessing intuitive understanding of material science and structural stability; (ii) Construction and Fabrication: evaluating logic of construction or fabrication of objects (e.g. joints, furniture) and small to large spatial structures; (iii) Vernacular Living: examining the knowledge of unique living style of local societies; and (iv) Cultural Context: testing model's understanding of regional, historic, and religious cultural representation.

**Environment Interaction**   The environment interaction category evaluates MLLMs' ability to compare options, make decisions and predict affordances in real-world environments, as a user or a decision maker of different spaces. User-Environment Interaction asks questions such as how to choose spaces under different weather conditions from a space user's perspective. Design-Environment Interaction asks questions from a space decision maker's view, for example, how to set up a space to achieve desired goals. Mobility in Space challenges the models to select or plan the navigation in various environments through different means of transportation or for different subjects (e.g. humans, vehicles, other animals). Finally, Environment Sustainability tests the understanding of models of environmental sustainability strategies and systems in the real world.

## 3.2   DATASET CONSTRUCTION

**Data curation**   The dataset collection consists of two steps. First, 51 university students representing multiple nationalities, cultural backgrounds and design traditions are asked to produce brand new diagrams for every subcategory. Students with design (mostly architecture-related) backgrounds are intentionally chosen because they are trained to possess strong spatial abilities, including the ability to create high-quality visual representations of the physical world from scratch. This approach yields two key benefits: (i) all diagrams are freshly created, eliminating data-contamination risks and preventing models from relying on prior exposure to publicly available diagrams; and (ii) by encouraging contributors to follow professional standards while retaining their personal drawing styles, we achieve greater visual diversity of the diagrams. Second, the contributors are asked to carefully craft two questions for every diagram, each with text or visual answer choices. Specific requirements are given to the contributors during the data generation process: (i) the diagrams must accurately represent information aligned with specific categories; (ii) the questions should be closely related to the diagram and the categories; (iii) reasoning process must be involved to answer the questions to avoid simple pattern match; (iv) linguistic or positional shortcuts should be avoided.

**Data format**   The type of question in the benchmark is visual question answering. All the questions are single-answer multiple-choice questions with four choices. However, while some questions have four text options, others have purely visual options, where question text contains only the labels of the visual options in the image. To mitigate the potential model bias on the answer's locations and labels, the choices of all questions are randomly shuffled so that the probability of each position (A, B, C and D) being the correct answer is approximately evenly distributed (25.46%, 25.37%, 25.46%, and 23.71%, respectively).

**Data quality control**   To further control the quality of the dataset, all 742 diagrams and 1484 questions produced then undergo a multi-stage refinement and screening process. Figure 3 illustrates the pipeline of data quality control.

(i) During the data creation phase, contributors met weekly with the meta-annotators to review a subset of their diagram–question pairs. In these meetings, ambiguous cases were discussed in detail, and concrete examples were illustrated. This iterative feedback loop ensured that contributors gradually converged to a shared understanding of the rubric, leading to de facto annotator agreement over time rather than idiosyncratic interpretations by different students.

(ii) Volunteers from various backgrounds representing the general population are invited to review all image-question pairs and point out clarity issues, logic flaws, and any other errors for the contributors to refine the diagrams and texts accordingly.

(iii) A few dedicated reviewing sessions were conducted with external reviewers who were not involved in the initial data creation. Their independent perspective helped to surface hidden biases or

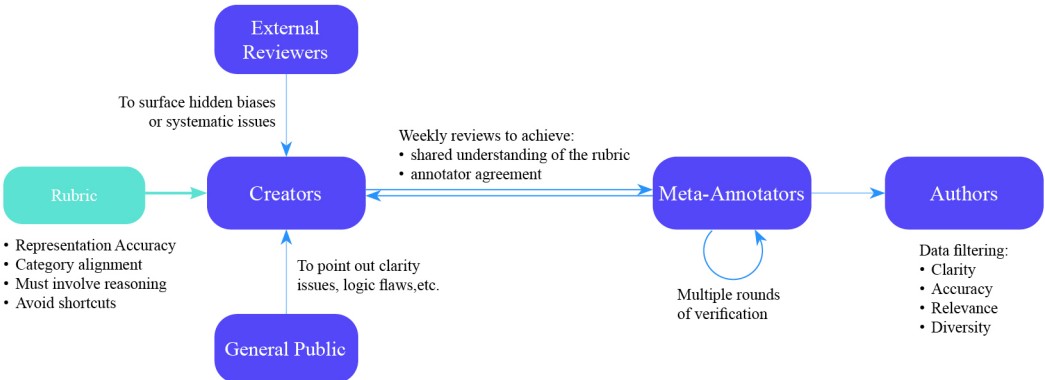

Figure 3: The pipeline of data quality control.

systematic issues that might not be obvious to the original creators, further increasing the soundness and robustness of the dataset.

(iv) Multiple rounds of verification are conducted by meta-annotators. At this stage, 50 questions are adversarially rewritten to eliminate linguistic shortcuts where applicable. For example, options with features such as special length and sentiment that make them appear more likely to be the correct answer are modified. This involves a small proportion of questions. Each data entry is examined to ensure that referring to the visual input is necessary to answer the questions. During this process, 1468 questions remain unchanged.

(v) Finally, all the data is filtered again by the authors, where the key selection criteria include clarity, accuracy, relevance, and diversity of the image-question pairs. In this process, 41 diagrams and 345 questions are excluded. As a result, 701 diagrams and 1139 questions that meet all requirements are kept.

## 3.3 DATA ANALYSIS

The dataset consists of 1139 image-question pairs in total, where Spatial Reasoning, Commonsense Knowledge and Environment Interaction contribute 400, 395 and 344 questions respectively. The proportion of each subcategories is displayed in Figure 2 (a).

In addition to the categories and subcategories, the dataset can be divided into nine groups according to the scale of space each question requires to be reasoned on, namely: object (e.g. furniture, columns, decorative tiles), room space (e.g. a bedroom, a hall, a unit/apartment), building space (an individual building or a building cluster, interior or exterior), spatial structure (e.g. a shell, a bridge, an arch), urban space (e.g. street blocks, road networks, a park, a waterfront, a village), abstract geometry and others. The distribution and example of each scale is illustrated in Figure 2 (b).

To further enhance models' visual reasoning capabilities and reduce reliance on purely linguistic cues, 40% of the questions are provided with purely visual options. To correctly answer these questions, the models need to not only perceive visual information of the question but also interpret the visual descriptions of the options or simulate corresponding views. Figure 1 Spatial Reasoning part and Figure 5 (b) demonstrate examples of questions with purely visual options. The rest 60% of questions provide text options.

To check the similarity of the diagrams to existing images, Google Cloud (2025) Vision is deployed to search for similar images of each diagram. An average cosine similarity is calculated between the diagrams and their top most similar images found in the search using CLIP (Radford et al., 2021) embeddings. The comparison reveals that 75% of the diagrams achieve similarity scores under 0.723, with 50% scoring below 0.665, and an overall mean of 0.654.

## 4 EXPERIMENTS

Based on the SpaCE-Eval benchmark, we conduct extensive evaluations on a selection of both proprietary and open-source MLLMs and analyse the results.

### 4.1 MODELS

We consider a wide range of proprietary and competitive open-source MLLMs to perform a comprehensive evaluation on SpaCE-Eval. For proprietary models, we consider GPT-5 (OpenAI, 2025), GPT-5-mini (OpenAI, 2025), GPT-4o (Hurst et al., 2024), GPT-4o-mini (Hurst et al., 2024),GPT-o4-mini (Hurst et al., 2024), grok-2-vision-1212 (xAI, 2024) (grok-2-vision), claude-3.7-sonnet (Anthropic, 2024), claude-sonnet-4 (Anthropic, 2025) and gemini-2.5-flash-preview (Gemini Team Google et al., 2023) (gemini-2.5-flash). For open-source models, we evaluate the Llama-4 (Meta AI, 2025) family, including Llama-4-Maverick-17B-128E-Instruct (Llama-4-Maverick) and Llama-4-Scout-17B-16E-Instruct (Llama-4-Scout); the gemma-3 (Gemma Team et al., 2025) family, including gemma-3-27b-it, gemma-3-12b-it and gemma-3-4b-it; the Qwen2.5-VL (Bai et al., 2025) family, including Qwen2.5-VL-72B-Instruct (Qwen2.5-VL-72B) and Qwen2.5-VL-7B-Instruct (Qwen2.5-VL-7B); and other representative MLLMs including Pixtral-12B (Agrawal et al., 2024), glm-4.1v-9b-thinking (GLM-V Team et al., 2025) (glm-4.1v-9b), Idefics3-8B-Llama3 (Laurençon et al., 2024), llava-onevision-7b (Li et al., 2024a), Phi-4-multimodal-instruct (Abouelenin et al., 2025) (Phi-4-multimodal) and smolvlm-2b (Marafioti et al., 2024). In addition, human volunteers of various backgrounds are invited to manually solve the questions, through which average human performance (Human Avg.) is obtained.

We evaluate most models using the OpenRouter API (OpenRouter, 2025) for efficiency, for models that are not supported by OpenRouter, we deploy the models using VLLM (Kwon et al., 2023) and use their default hyperparameter for inference. The detailed prompt structure and model API or links are provided in Appendix A.0.2. When the model prediction is not exactly the same expression as the ground truth (e.g. model prediction is not answer "A", but may have the same linguistic meaning as option A), we use GPT-4o-mini (Hurst et al., 2024) to classify whether the prediction is correct.

### 4.2 MAIN RESULTS

In this section, we present a comprehensive comparison of different MLLMs based on SpaCE-Eval, the details are shown in Table 2. For each model, the accuracy is represented by the percentage of correct predictions out of the total predictions in each category. We summarise our key finding as follows.

**Overall result**  SpaCE-Eval is a very challenging benchmark. The best result across all the tested models is only 56.37% achieved by GPT-5 (OpenAI, 2025), The proprietary models achieve an overall accuracy between 39.68% and 56.37%, and the majority of open-source models can only reach an overall accuracy of less than 40%, except for Llama-4-Maverick (Meta AI, 2025), glm-4.1v-9b (GLM-V Team et al., 2025)and llava-onevision-7b (Li et al., 2024a), whose accuracy is 45.92%, 43.37% and 42.41% respectively.

**Cross-category result**  Trained on large-scale data, the models achieve as highest as 66.08% and 61.63% in the Commonsense Knowledge and Environment Interaction categories. However, their performance in the Spatial Reasoning is dramatically lower, with scores of only 42.25%, with the majority of which only slightly higher than random guesses, highlighting a significant capability gap between Spatial Reasoning and the other two categories.

**Cross-model-scale result**  According to the results, generally, the models with larger sizes perform better than those with smaller sizes. For example, in the Gemma (Gemma Team et al., 2025) series, the largest model (gemma-3-27b-it) has the best performance (39.77%), the middle size version (gemma-3-12b-it) has the medium accuracy (37.05%), and the smallest model (gemma-3-4b-it ) has the least accuracy (36.00%). Similarly, Qwen2.5-VL-72B-Instruct (Bai et al., 2025) has higher overall accuracy than Qwen2.5-VL-7B-Instruct.

Table 2: Evaluation results of various MLLMs on SpaCE-Eval. We evaluate the three main categories: Spatial Reasoning, Commonsense Knowledge and Environment Interaction. For each main category, we evaluate the four subcategories. SI, SA, VA and FT represent Spatial Interpretation, Space Association, View Analysis and Form Transformation, respectively. MaS, CF, CC and VL represent Materials and Structures, Construction and Fabrication, Cultural Context and Vernacular Living, respectively. UEI, DEI, MiS and ES represent User-Environment Interaction, Design-Environment Interaction, Mobility in Space and Environment Sustainability, respectively. **Bold** indicates the best performance of each category, while underlined denote the second-best performance in each category.

| Model Name | Overall Mean | Spatial Reasoning | | | | | Commonsense Knowledge | | | | | Environment Interaction | | | | |
|---|---|---|---|---|---|---|---|---|---|---|---|---|---|---|---|---|
| | | SI | SA | VA | FT | Mean | MaS | CF | CC | VL | Mean | UEI | DEI | MiS | ES | Mean |
| Human Avg. | 79.00 | 84.92 | 89.19 | 92.24 | 76.24 | 84.18 | 81.48 | 65.32 | 58.2 | 81.86 | 71.83 | 72.64 | 80.82 | 91.84 | 80.17 | 81.34 |
| **Proprietary MLLMs** | | | | | | | | | | | | | | | | |
| GPT-5 | **56.37** | **39.70** | **45.95** | **46.55** | **43.4** | **42.25** | 69.44 | 61.29 | **62.60** | **69.61** | **66.08** | 65.79 | 60.27 | 50.0 | 74.58 | **61.63** |
| GPT-5-mini | 52.15 | 33.67 | **45.95** | 39.66 | 38.68 | 37.00 | **70.37** | 58.06 | 53.66 | 62.75 | 61.27 | 58.77 | 54.79 | 53.06 | 76.27 | 59.3 |
| GPT-4o-mini | 39.68 | 16.08 | 27.03 | 32.76 | 29.25 | 23.00 | 62.96 | 45.16 | 36.59 | 47.06 | 47.85 | 55.26 | 45.21 | 36.73 | 66.10 | 49.71 |
| GPT-o4-mini | 45.04 | 25.63 | 29.73 | 37.93 | 25.47 | 27.75 | 61.11 | 56.45 | 47.51 | 56.86 | 54.94 | 56.14 | 42.47 | 47.96 | 72.88 | 53.78 |
| grok-2-vision | 42.32 | 30.15 | 24.32 | 32.76 | 30.19 | 30.00 | 59.26 | 45.16 | 35.77 | 54.90 | 48.61 | 55.26 | 35.62 | 41.84 | 67.80 | 49.42 |
| claude-sonnet-4 | 48.64 | 29.65 | 37.84 | 41.38 | 28.30 | 31.75 | 64.81 | 53.23 | 53.66 | 63.73 | 59.24 | 57.89 | 49.32 | 50.00 | 71.19 | 56.10 |
| claude-3.7-sonnet | 47.41 | 26.63 | 16.22 | 41.38 | 33.96 | 29.75 | 65.74 | 53.23 | 46.34 | 63.73 | 57.22 | 54.39 | 47.95 | 56.12 | 72.88 | 56.69 |
| gemini-2.5-flash | 47.50 | 25.13 | 24.32 | 43.10 | 28.30 | 28.50 | 64.81 | **62.90** | 47.97 | 58.82 | 57.72 | 61.40 | 49.32 | 51.02 | 72.88 | 57.85 |
| **Open-source MLLMs** | | | | | | | | | | | | | | | | |
| Qwen2.5-VL-72B | 37.84 | 19.10 | 35.14 | 27.59 | 33.96 | 25.75 | 57.41 | 38.71 | 26.83 | **59.80** | 45.57 | 53.51 | 35.62 | 31.63 | 50.85 | 43.02 |
| gemma-3-27b-it | 39.77 | 23.12 | 21.62 | 39.66 | 21.70 | 25.00 | 57.41 | **58.06** | 32.52 | 55.88 | 49.37 | 46.49 | **52.05** | 34.69 | 55.93 | 45.93 |
| Llama-4-Maverick | **45.92** | 22.11 | 37.84 | 37.93 | 29.25 | 27.75 | **67.59** | **58.06** | **47.97** | 49.02 | **55.19** | 57.89 | 49.32 | **44.90** | 81.36 | **56.40** |
| Llama-4-Scout | 39.42 | 18.09 | 21.62 | 34.48 | 29.25 | 23.75 | 56.48 | 48.39 | 36.59 | 46.08 | 46.33 | 52.63 | 42.47 | 41.84 | 66.10 | 49.71 |
| gemma-3-12b-it | 37.05 | 19.10 | 16.22 | 36.21 | 22.64 | 22.25 | 52.78 | 45.16 | 33.33 | 50.00 | 44.81 | 42.11 | 39.73 | 35.71 | 74.58 | 45.35 |
| Pixtral-12B-2409 | 39.77 | 25.13 | 27.03 | 37.93 | 33.96 | 29.50 | 50.93 | 45.16 | 34.96 | 48.04 | 44.30 | 49.12 | 39.73 | 30.61 | 76.27 | 46.51 |
| glm-4.1v-9b | 43.37 | 24.12 | 37.84 | 36.21 | 29.25 | 28.50 | 60.19 | 41.94 | 47.97 | 50.00 | 50.89 | 50.88 | 50.68 | 42.86 | 71.19 | 52.03 |
| Idefics3-8B-Llama3 | 38.72 | 21.11 | 18.92 | **44.83** | 34.91 | 28.00 | 56.48 | 41.94 | 35.77 | 49.02 | 45.82 | 54.39 | 35.62 | 29.59 | 52.54 | 43.02 |
| Qwen2.5-VL-7B | 37.14 | 19.60 | 21.62 | 41.38 | 31.13 | 26.00 | 48.15 | 45.16 | 34.96 | 41.18 | 41.77 | 47.37 | 41.10 | 34.69 | 61.02 | 44.77 |
| llava-onevision-7b | 42.41 | 26.63 | 35.14 | 37.93 | 33.96 | **31.00** | 58.33 | 56.45 | 37.40 | 50.98 | 49.62 | 56.14 | 41.10 | 31.63 | 64.41 | 47.38 |
| Phi-4-multimodal | 38.72 | 27.14 | 29.73 | 31.03 | **35.85** | 30.25 | 49.07 | 30.65 | 36.59 | 41.18 | 40.25 | 45.61 | 47.95 | 37.76 | 62.71 | 46.80 |
| gemma-3-4b-it | 36.00 | 22.11 | 24.32 | 32.76 | 24.53 | 24.50 | 55.56 | 35.48 | 30.89 | 45.10 | 42.03 | 43.86 | 31.51 | 39.80 | 57.63 | 42.44 |
| smolvlm-2b | 36.00 | **28.14** | 32.43 | 27.59 | 33.96 | 30.00 | 58.33 | 41.94 | 34.96 | 35.29 | 42.53 | 38.60 | 27.40 | 33.67 | 42.37 | 35.47 |

**Human vs. model** In contrast to model performances, the average human results in Table 2 demonstrate a different pattern. Humans achieve the highest results in Spatial Reasoning and the second highest in Environmental Interaction. Both categories require stronger reasoning abilities than the Commonsense Knowledge category, especially for Spatial Reasoning. Noticeably, GPT-5 (OpenAI, 2025) has surpassed human average results in the Cultural Context subcategory, under Commonsense Knowledge. Despite such differences, human performance in spatial reasoning (84.18%) and environment interaction (81.34%) is significantly higher than all the models (at best 42.25% and 61.63% respectively). This result suggests that humans are comparatively stronger in reasoning than in memorisation, and maintain a clear advantage over current models in reasoning ability.

**Visual vs. textual** The evaluation reveals notable imbalance between textual and visual reasoning. Figure 4(a) compares model performance on questions with textual options and those with purely visual options. Consistently, throughout all categories, the models perform significantly lower in visual-only options compared to options with text. This disparity indicates that MLLMs possess substantially weaker capabilities in visual comprehension and reasoning compared to textual reasoning. Additionally, Figure 4(b) further compares performance with and without image input. When images

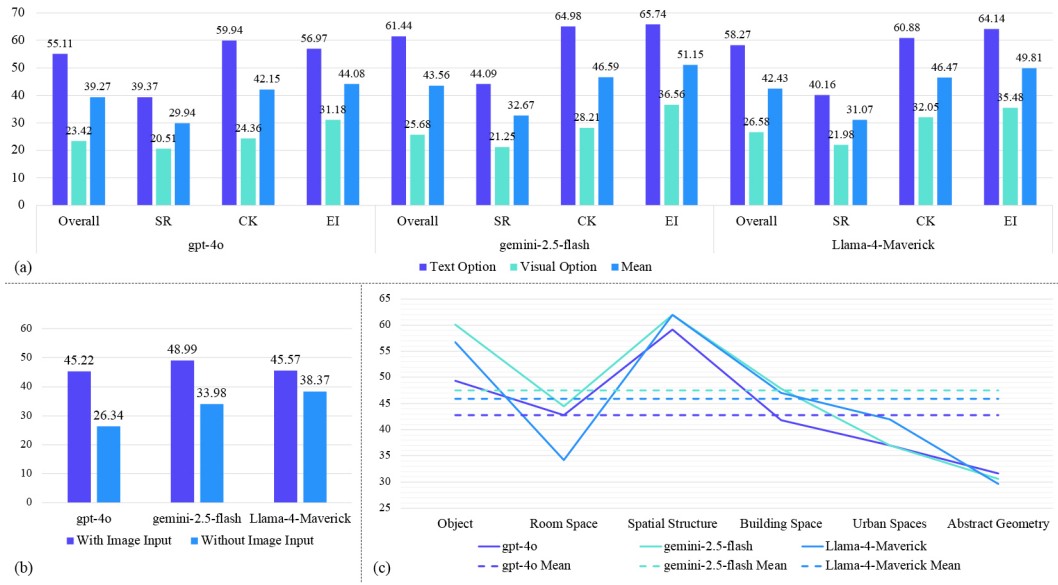

Figure 4: (a) Model performances on questions with textual options and those with purely visual options. (b) Model performances on questions with and without image input. (c) Models' accuracy across different scales. The dashed lines indicate models' mean accuracy over all scales.

are absent, model accuracy declines significantly, underscoring the strong dependency of MLLMs on visual inputs to answer questions in our benchmark.

## 4.3  Analysis of Current Limitations

A particularly prominent limitation revealed by the evaluation is the models' incapacity to bridge spatial scales, especially when reasoning across local-to-global and global-to-local relations. This may stem from challenges in reasoning over larger spatial extents and in performing spatial simulation. In addition, the models exhibit difficulties in abstract interpretation and reasoning. We briefly analyse these failure patterns in this section. More in-depth diagnostic analysis are provided in A.0.4, where we detail how models arrive at incorrect answers and discuss the likely underlying causes of these failures.

**Difficulty in reasoning on larger spatial scales**  As shown in Figure 4(c), the models exhibit a consistent trend across different spatial scales: they achieve relatively strong performance in object-scale reasoning, but performance declines as the scale expands to rooms, buildings, and urban spaces, with abstract geometry reasoning posing the greatest challenge. In other words, models demonstrate weaker understanding and reasoning capabilities for larger-scale spaces compared to object-level reasoning, and their performance in these cases falls below the overall mean across all scales. This trend can be resulted from the fact that many existing datasets primarily focus on object-level reasoning, as noted in Section 1, highlighting the need for datasets and benchmarks that target larger spatial scales. By contrast, performance on spatial-structure tasks is slightly higher, as these tasks emphasise intuitive understanding of structural patterns rather than the more demanding reasoning processes required in the other scale groups.

**Difficulty in spatial simulation**  Models demonstrate critical shortcomings when spatial simulation is required in the tasks. This limitation often leads to failures in questions involving global–local relationships. The evaluation shows that models frequently struggle to identify the view corresponding to a given location, or to localise the position associated with a given view. Typical questions of this type include: (i) given a global view, what would be observed when standing at position A and looking in the direction of the arrow? or (ii) given a local view, at which position in the room, building, or village would this perspective be obtained? Figure 5(a) presents an example of such

location–view association tasks, which are usually posed with pure visual options. Across models, the accuracy on these questions remains low, with fewer than 22.73% of responses being correct. Such failures commonly arise when models are unable to mentally simulate or infer the consequences of spatial transformations, changes in perspective, or dynamic interactions within the environment. This highlights a fundamental weakness of MLLMs in the spatial reasoning abilities essential for real-world understanding.

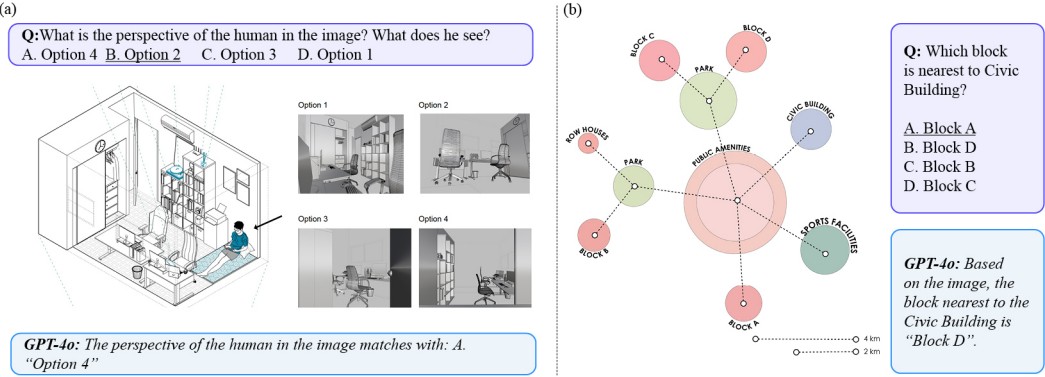

Figure 5: (a) An example of location-view association. (b) An example of abstract representation of distance. The ground truth is underlined.

**Difficulty in abstract interpretation and reasoning** When challenged with questions involving spatial measurements, models often resort to comparing pixel-level distances rather than reasoning with the abstract spatial relations encoded in the diagrams. As shown in Figure 5(b), GPT-4o (Hurst et al., 2024) bases its answer on the shorter pixel distance between the circles representing the Civic Building and Block D, while disregarding the abstract distance indicated by dashed lines, which reflect the actual spatial relationship in the real world. This illustrates a broader limitation: the models tend to privilege surface-level visual patterns over the abstract principles that govern spatial and physical phenomena. Such behavior raises concerns about their reliability in tasks where correct interpretation requires abstraction beyond visual similarity, underscoring the gap between current multimodal reasoning and human-like spatial understanding.

## 5 CONCLUSION

As existing benchmarks do not comprehensively assess the ability of MLLMs to understand and reason in real-world contexts, we introduce SpaCE-Eval, a meticulously constructed benchmark designed to evaluate several crucial aspects of real-world understanding. The benchmark undergoes a rigorous preparation and selection process, followed by extensive experiments on a range of state-of-the-art MLLMs. The results reveal that while current models approach human performance in knowledge-intensive categories, they fall considerably short in reasoning-intensive ones, particularly in spatial reasoning. Key limitations include difficulties in bridging spatial scales, performing spatial simulation, and engaging in abstract reasoning, underscoring the need for continued advancements in both benchmark datasets and MLLM architectures.

## ACKNOWLEDGMENTS

This research is supported by the Hokkien Foundation Early Career Professorship through the Artificial-Architecture Lab. This research is supported by the Ministry of Education, Singapore, under its Academic Research Fund (AcRF) Tier 1 grant, and funded through the SUTD Assistant Professorship Scheme (SAP 2025_001). We thank the students of 20.224 Artificial & Architectural Intelligence in Design class (2025) at the Singapore University of Technology and Design for their contributions to the raw data.

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

# A  APPENDIX

## A.0.1  DATASET CATEGORY DETAILS

This section demonstrates examples of each subcategory of the four main categories in the data set.

**Spatial Reasoning**

Subcategory: Spatial Interpretation

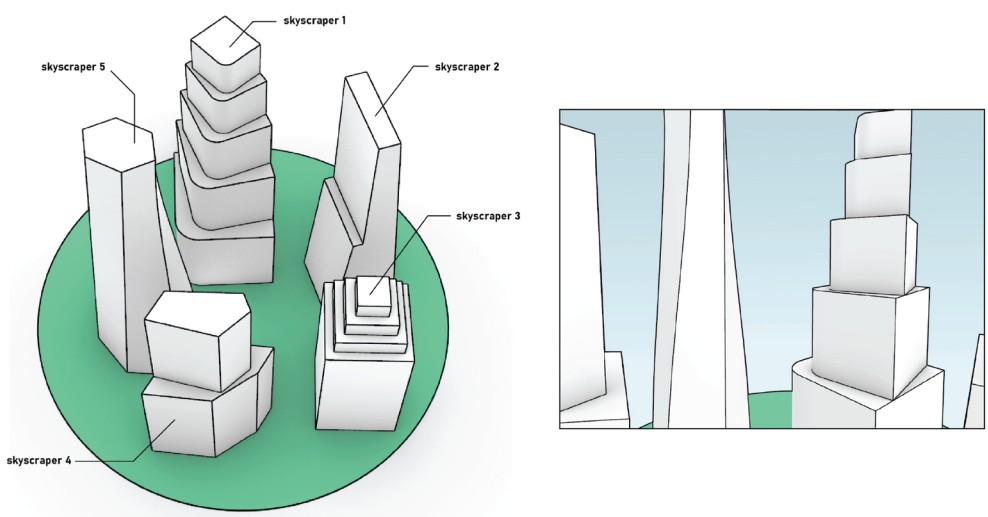

**Question**

From which skyscraper was the perspective view shown on the right taken from?

A. Skyscraper 3      B. Skyscraper 4
C. Skyscraper 5      D. Skyscraper 2

**Ground truth**

A. Skyscraper 3

**GPT-4o prediction**

*The perspective view on the right was taken from Skyscraper 2. Correct answer: D. Skyscraper 2*

☒

Figure 6: Example of Spatial Reasoning/Spatial Interpretation.

**Spatial Reasoning**

Subcategory: Space Association

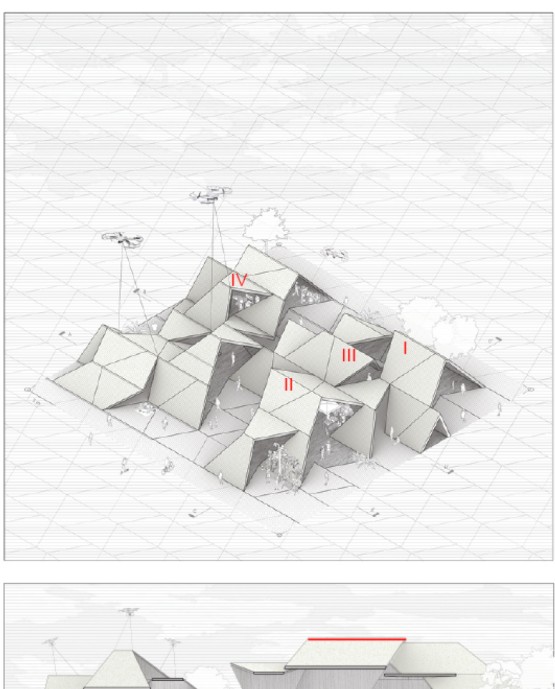

**Question**

Where is the highlighted roof (in the section diagram) located on the axonometric drawing?

A. II      B. III      C. IV      D. I

**Ground truth**

B. III

**GPT-4o prediction**

*The highlighted roof in the section diagram corresponds to location D (I) on the axonometric drawing.*

☒

Figure 7: Example of Spatial Reasoning/Space Association.

**Spatial Reasoning**

Subcategory: View Analysis

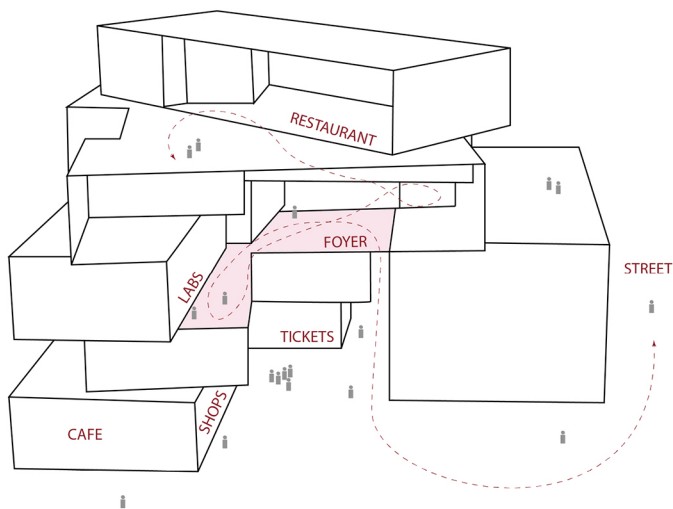

**Question**

If I am standing in front of the foyer, most likely which space is outside of my view?

A. none of the spaces      B. cafe      C. shops      D. labs

**Ground truth**

B. cafe

**GPT-4o prediction**

*D. labs*

☒

Figure 8: Example of Spatial Reasoning/View Analysis.

**Spatial Reasoning**

Subcategory: Form Transformation

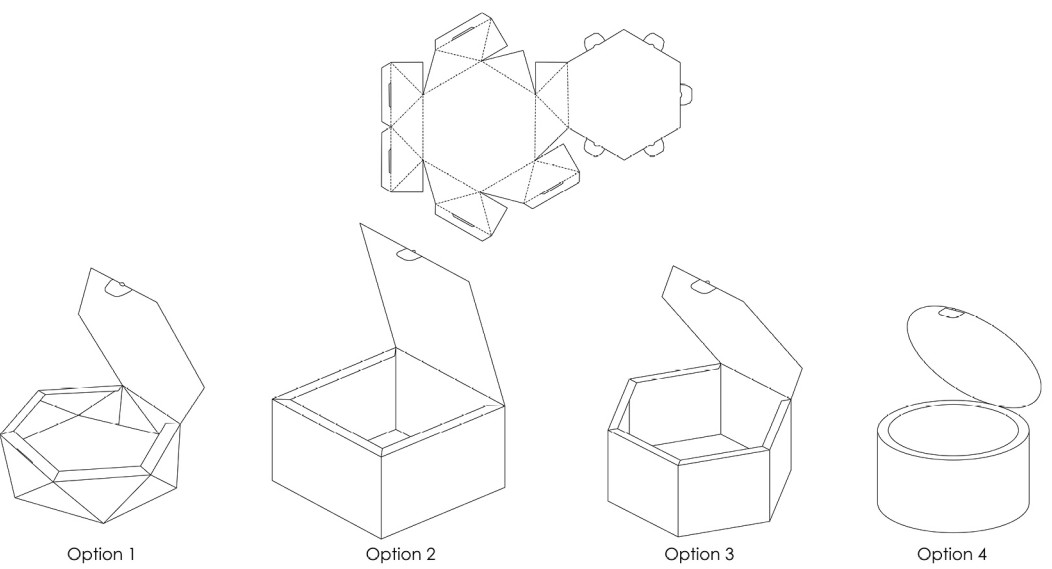

**Question**

Given is a flat folding plan drawing of an origami box, what is the correct box this can produce?

A. Option 3     B. Option 1     C. Option 4     D. Option 2

**Ground truth**

B. Option 1

**GPT-4o prediction**

*The correct answer is A. Option 3*

☒

Figure 9: Example of Spatial Reasoning/Form Transformation.

**Commonsense Knowledge**

Subcategory: Materials and Structures

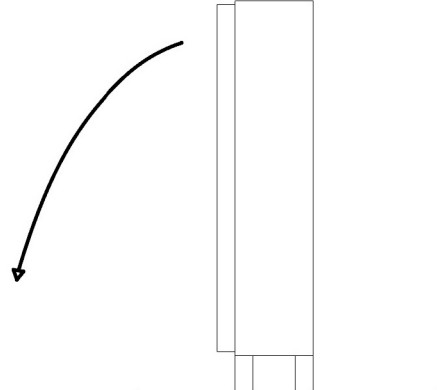

**Question**

The wardrobe drawn above keeps toppling in the direction indicated by the arrow. What is the most likely reason?

A. The legs of the wardrobe have different lengths
B. The doors are heavier than the rest of the wardrobe
C. It is not placed against a wall
D. Items inside are places too close to the door

**Ground truth**

B. The doors are heavier than the rest of the wardrobe

**GPT-4o prediction**

*Based on the image, the most likely reason the wardrobe is toppling in the direction indicated by the arrow is: A. The legs of the wardrobe have different lengths. This would cause an imbalance and lead to toppling in one direction.*

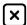

Figure 10: Example of Commonsense Knowledge/Materials and Structures.

**Commonsense Knowledge**

Subcategory: Construction and Fabrication

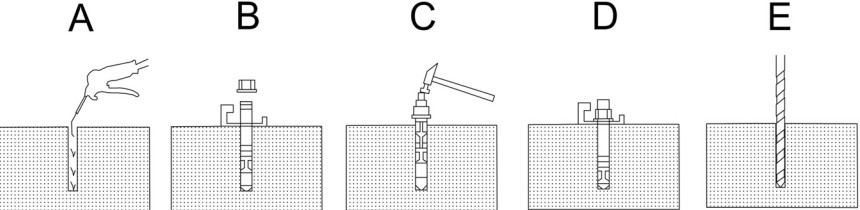

**Question**

What is the correct sequence to install a fastener in cement and concrete

A. EACBD
B. AECDB
C. EACDB
D. AECBD

**Ground truth**

A. EACBD

**GPT-4o prediction**

*To determine the correct sequence, let's analyze the sequence of steps visualized: 1. E - Drilling a hole in the concrete. 2. A - Clearing the hole of debris. 3. C - Inserting the fastener. 4. D - Setting the fastener with a tool. 5. B - Placing and tightening the nut or washer. The correct sequence is EACDB. Answer: C. EACDB*

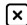

Figure 11: Example of Commonsense Knowledge/Construction and Fabrication.

**Commonsense Knowledge**

Subcategory: Cultural Context

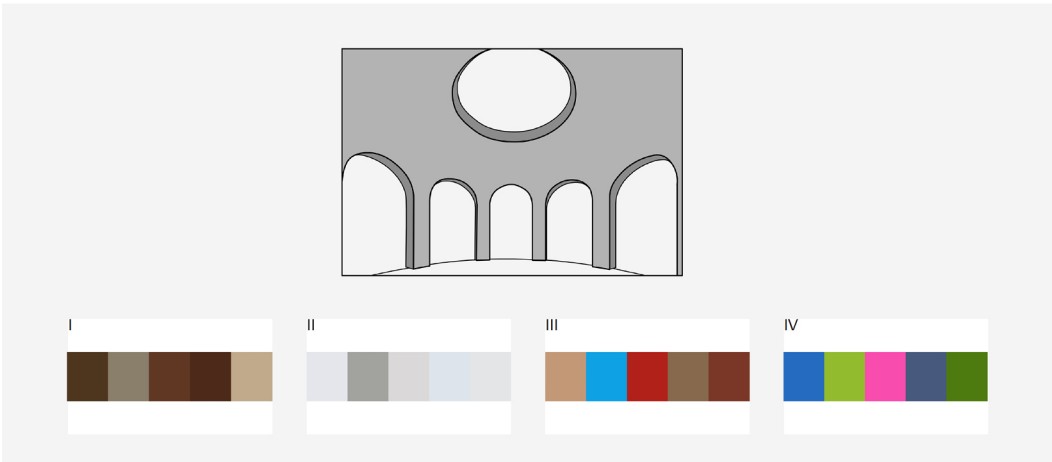

**Question**

Which color palette is best appropriate if the architecture in the diagram is influenced by Classical Mediterranean architecture?

A. I     B. II     C. III     D. IV
**Ground truth**

B. II

**GPT-4o prediction**

*The color palette best appropriate for architecture influenced by Classical Mediterranean architecture is: A. I This palette contains earthy tones typical of the Mediterranean region.*

☒

Figure 12: Example of Commonsense Knowledge/Cultural context.

**Commonsense Knowledge**

Subcategory: Vernacular Living

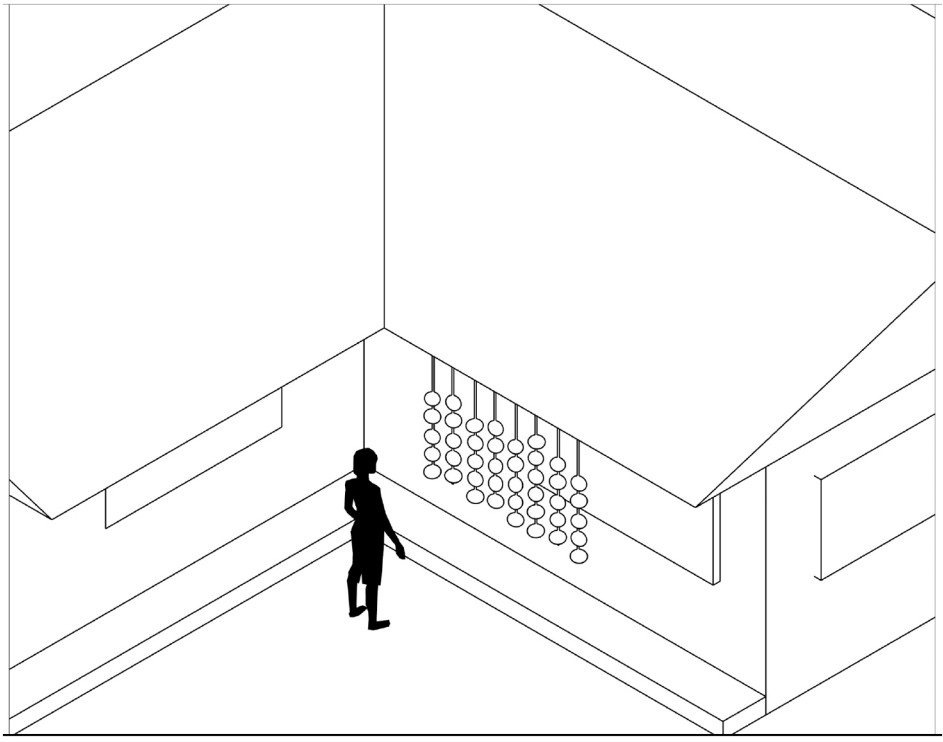

**Question**

There are objects hanging from the roof of the Korean house. What are the objects?

A. Laundry        B. Wind chimes      C. Fruit put out to dry D. Ornamental beads

**Ground truth**

C. Fruit put out to dry

**GPT-4o prediction**

*The objects hanging from the roof of the Korean house appear to be C. Fruit put out to dry.*

☑

Figure 13: Example of Commonsense Knowledge/Vernacular Living.

**Environment Interaction**

Subcategory: User-Environment Interaction

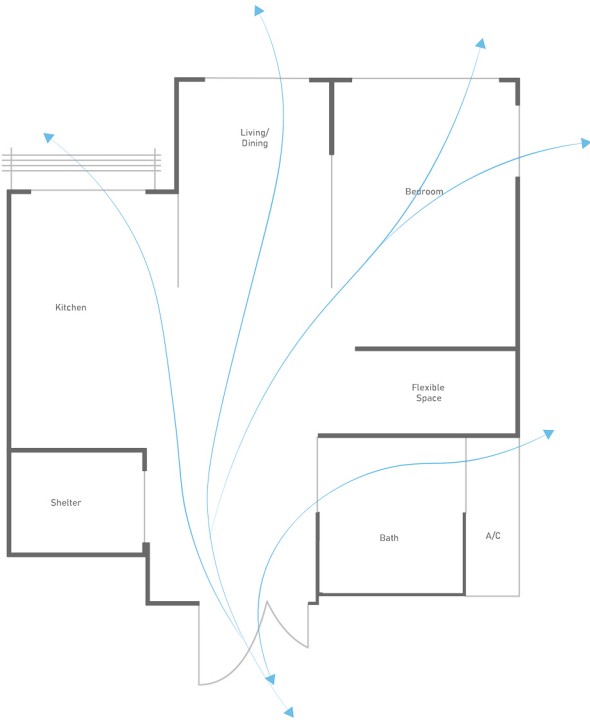

**Question**

From the diagram, please select the spaces least suitable for long-term comfort as a result of being stagnant-air zones.

A. Living and Dining Space
B. Kitchen
C. Flexible Space
D. Bath

**Ground truth**

C. Flexible Space

**GPT-4o prediction**

*The space least suitable for long-term comfort due to being a stagnant-air zone is the "Flexible Space".*

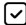

Figure 14: Example of Environment Interaction/User-Environment Interaction.

**Environment Interaction**

Subcategory: Design-Environment Interaction

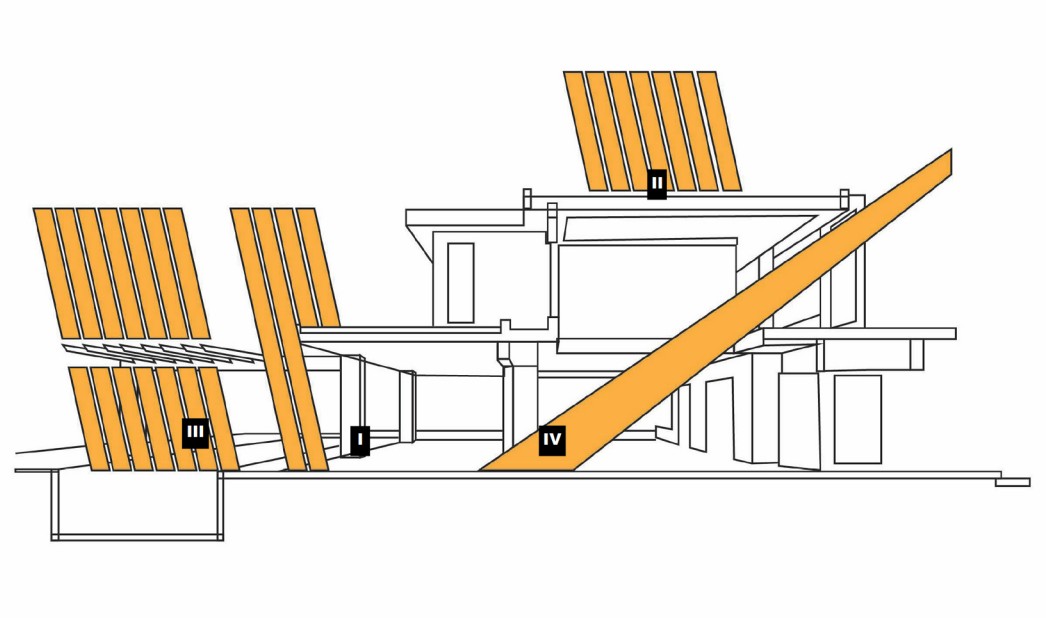

**Question**

Which position would be the most ideal for photovoltaic panels to be installed. Consider maintenance and ease of access.

A. III      B. III      C. II      D. IV

**Ground truth**

C. II

**GPT-4o prediction**

*Based on the image and considering maintenance and ease of access, the most ideal position for photovoltaic panels would be: A. III*

❌

Figure 15: Example of Environment Interaction/Design-Environment Interaction.

**Environment Interaction**

Subcategory: Mobility in Space

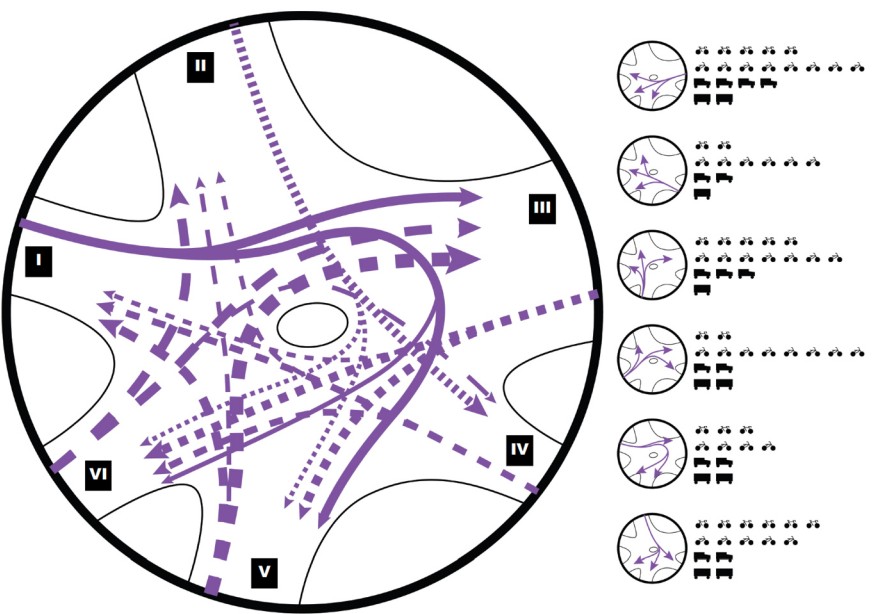

**Question**

Which point of the roundabout has the highest volume flow entering the roundabout?

A. IV      B. III      C. V      D. II

**Ground truth**

B. III

**GPT-4o prediction**

*The point with the highest volume flow entering the roundabout is point "II".*

☒

Figure 16: Example of Environment Interaction/Mobility in Space.

**Environment Interaction**

Subcategory: Environment Sustainability

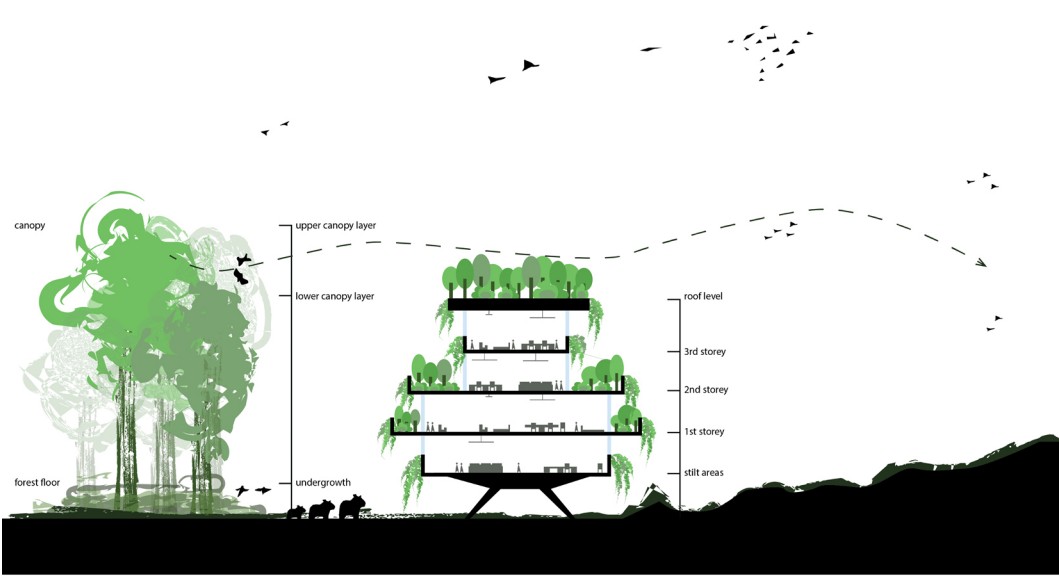

**Question**

Why is the building on stilts?

A. To allow cyclists to traverse freely on ground.
B. To mitigate flooding.
C. To allow ground animals to roam freely with minimal obstruction.
D. To allow visitors to look at animals from above freely."

**Ground truth**

C. To allow ground animals to roam freely with minimal obstruction.

GPT-4o prediction

*B. To mitigate flooding.*

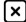

Figure 17: Example of Environment Interaction/Environment Sustainability.

### A.0.2  EXPERIMENT DETAILS

This section explains additional details of the experiment.

Prompt

For each data case, the prompt includes an image and the text of the question formatted as illustrated in Figure 18 (a). For questions with visual options, the options and the main diagram are prompted as one image as shown in Figure 18 (b). The prompt for questions with text options is shown in Figure 18 (c).

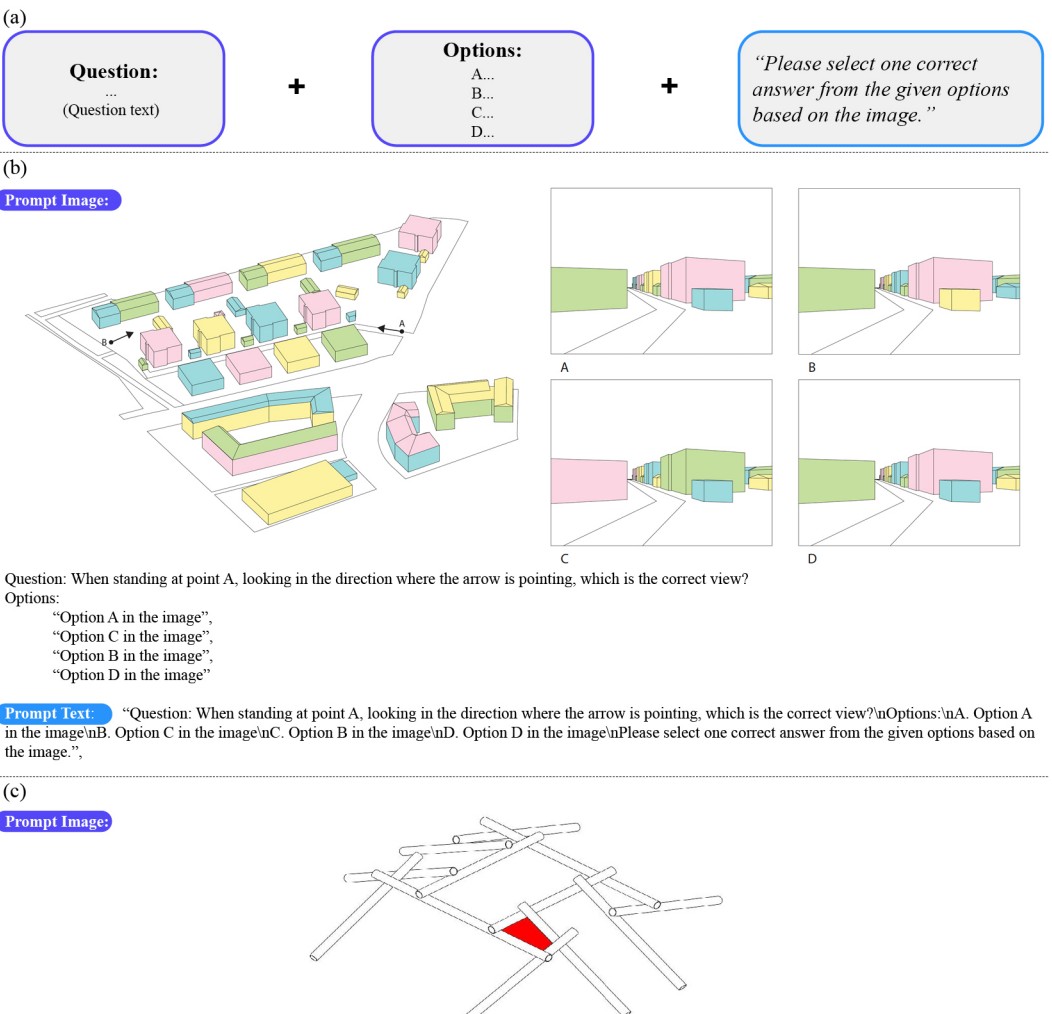

Figure 18: Prompt example used in the experiment, (a) structure of the prompt text. (b) prompt example of questions with visual options. (c) prompt example of questions with text options.

Model Links

We provide all links to the APIs or models used in our experiments in Table 3.

Table 3: Links to the models in our experiment

| Model | API/Model Link |
|---|---|
| GPT-5 | https://openrouter.ai/openai/gpt-5/api |
| GPT-5-mini | https://openrouter.ai/openai/gpt-5-mini/api |
| GPT-4o | https://openrouter.ai/openai/gpt-4.1-nano |
| GPT-4o-mini | https://openrouter.ai/openai/gpt-4o-mini |
| GPT-o4-mini | https://openrouter.ai/openai/gpt-4o-mini/api |
| grok-2-vision | https://openrouter.ai/x-ai/grok-2-vision-1212 |
| claude-sonnet-4 | https://openrouter.ai/anthropic/claude-sonnet-4/api |
| claude-3.7-sonnet | https://openrouter.ai/anthropic/claude-3.7-sonnet |
| gemini-2.5-flash | https://openrouter.ai/google/gemini-2.5-flash-preview/api |
| Qwen2.5-VL-72B | https://openrouter.ai/qwen/qwen2.5-vl-72b-instruct/api |
| gemma-3-27b-it | https://openrouter.ai/google/gemma-3-27b-it/api |
| Llama-4-Maverick | https://openrouter.ai/meta-llama/llama-4-maverick/api |
| Llama-4-Scout | https://openrouter.ai/meta-llama/llama-4-scout/api |
| gemma-3-12b-it | https://openrouter.ai/google/gemma-3-12b-it/api |
| Pixtral-12B-2409 | https://openrouter.ai/mistralai/pixtral-12b/api |
| glm-4.1v-9b-thinking | https://openrouter.ai/thudm/glm-4.1v-9b-thinking/api |
| Idefics3-8B-Llama3 | https://huggingface.co/HuggingFaceM4/Idefics3-8B-Llama3 |
| Qwen2.5-VL-7B | https://openrouter.ai/qwen/qwen-2.5-vl-7b-instruct/api |
| llava-onevision-7b | https://huggingface.co/llava-hf/llava-onevision-qwen2-7b-ov-hf |
| Phi-4-multimodal | https://openrouter.ai/microsoft/phi-4-multimodal-instruct/api |
| gemma-3-4b-it | https://openrouter.ai/google/gemma-3-4b-it/api |
| smolvlm-2b | https://huggingface.co/HuggingFaceTB/SmolVLM2-2.2B-Instruct |

### A.0.3 NEGATIVE DATA EXAMPLES

This section shows examples of diagram–question pairs removed for failing quality control with explanations.

**Visual Independence**

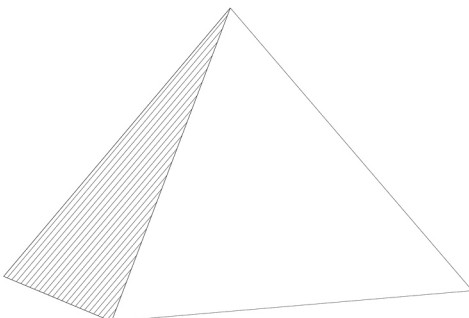

Which of the following pyramids is known for its significant role in shaping the ancient Egyptian view of the afterlife, and what period did its construction reflect in terms of architectural and religious evolution?
A. The Pyramid of Khufu, around 2600 BCE
B. The Pyramid of Khafre, around 2500 BCE
C. The Step Pyramid of Djoser, around 2630 BCE
D. The Bent Pyramid, around 2570 BCE

*Explanation: The question can be answered without visual input. The question can be fully answered based on historical knowledge of Egyptian monuments and does not rely on interpreting the accompanying image.*

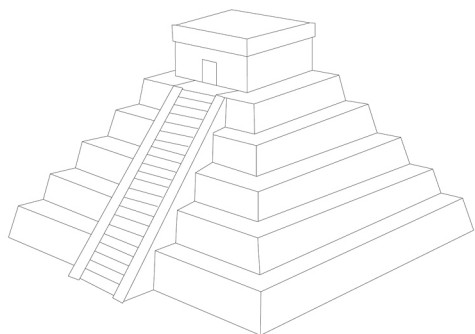

Which ancient civilization is most closely associated with the construction of monumental stepped pyramids, and what was the primary purpose of these structures?
A. Ancient Egypt      B. Ancient Mesopotamia
C. Mesoamerican Civilizations    D. Ancient Greece

*Explanation: The question can be answered without visual input. The origins and purpose of stepped pyramids can be identified from general historical knowledge, independent of the visual provided.*

Figure 19: Examples of questions that do not require visual input to answer.

**Question Cannot be Answered**

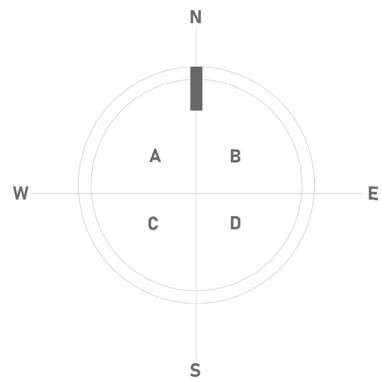

Which combination of quadrants are the most suitable for living spaces and offices?
A. A and B
B. A and C
C. B and C
D. A and D

*Explanation: The diagram does not provide enough information to answer the question. The diagram only shows compass directions and a simple division into four quadrants. There is no information about climate, sunlight orientation, prevailing winds, surrounding context, or functional requirements, without which it is impossible to determine which quadrants would be most suitable.*

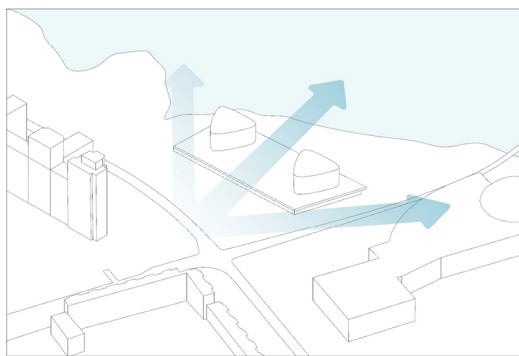

What do the arrows in the diagram represent?

A. Wind direction in relation to the building

B. Viewpoints towards the building

C. Pedestrian movement through the site

D. Access routes to and from the building

*Explanation: The diagram does not provide enough information to answer the question. The diagram shows arrows originating from an undefined area and pointing toward the blue area, but it does not include labels, a legend, or contextual cues. The graphic alone is insufficient to determine their specific meaning.*

Figure 20: Examples of diagrams that do not provide sufficient information to answer the questions.

**Lack of Clarity**

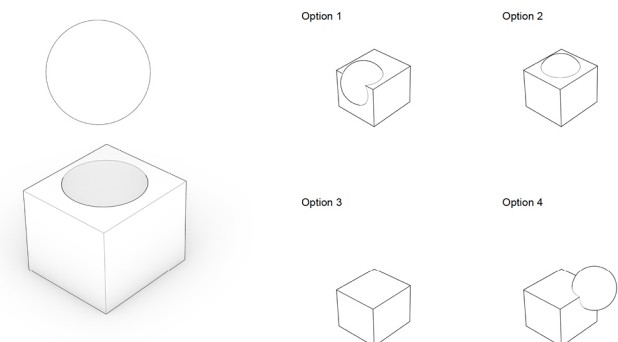

When the two forms on the left are assembled, which final 3D shape will they create?
A. Option 1.
B. Option 2.
C. Option 3.
D. Option 4.

*Explanation: The diagram does not provide enough visual information to determine whether the upper element is a flat circle or a sphere. Without clarity on its dimensionality or how it interacts spatially with the cube, it is impossible to identify which of the four configurations is correct.*

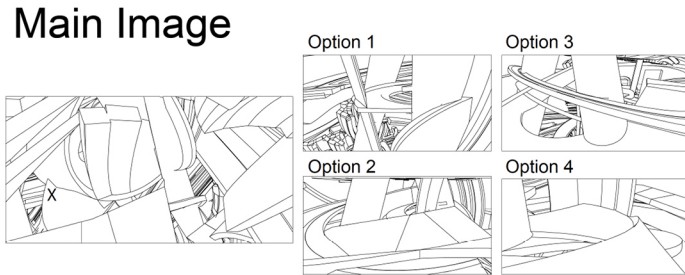

Which view will you see when standing at the point marked X in the Main Diagram?
A. Option 1 B. Option 2 C. Option 3 D. Option 4

*Explanation: The messy lines and curves make surfaces and volumes unclear. Without clarity in spatial relationships, it is hard to determine which of the four provided views corresponds to the position marked X.*

Figure 21: Examples of diagrams with insufficient visual clarity.

### A.0.4 DETAILED FAILURE CASE ANALYSIS

This section further demonstrates some typical failure cases, explaining how the model failed and what maybe the reasons causing the failure in detail.

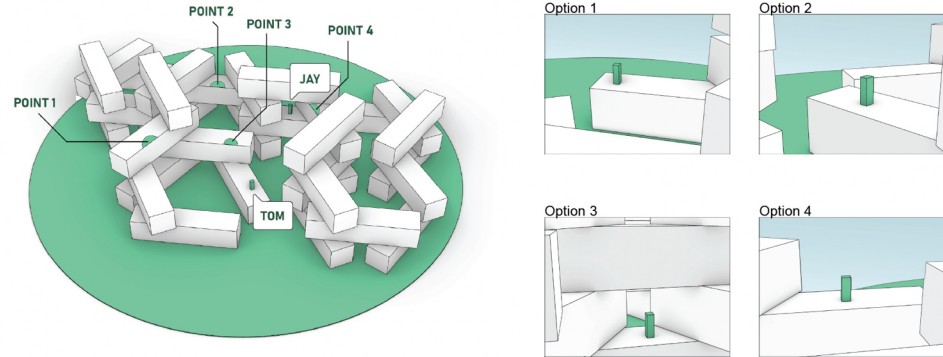

Which of the following perspectives correspond to how Jay will see Tom from his location?
A. Option 1 in the image        B. Option 2 in the image ☑
C. Option 3 in the image        D. Option 4 in the image

**GPT-5 Prediction**

*Jay's location and viewing direction are clearly shown in the left diagram:*
*Jay is positioned inside a cluster of tall block-like walls.*
*Tom is farther down and slightly to Jay's left, but Jay's view toward Tom is strongly obstructed by multiple walls.*
*Only a small gap allows Jay to see Tom, and the view would be:*
*Framed tightly by walls, almost like looking through a slot.*
*Tom appears low in the frame with walls very close on both sides.*
*Now compare the options:*
***Option 1***
*Too open. Jay would not see this much open ground; his view is blocked on both sides. Incorrect*
***Option 2***
*Still too open on the right. Jay is surrounded by higher and closer walls than shown here. Incorrect*
***Option 3***
*Tom is seen through a narrow opening, with:*
*A large wall overhead,*
*Tall walls on both sides,*
*A restricted field of view.*
*Matches Jay's actual obstructed sightline*
***Option 4***
*Too clear and too open. Jay wouldn't have a wide clear view like this given the wall cluster around him. Incorrect*

**Analysis**

This is a failure of **scale-aware spatial reasoning** and **egocentric viewpoint transformation**. The model did not recognise that the white blocks are in building scale compared to the size of Jay and Tom, and the openings are actually large enough to permit visibility. It focused overly on the 2D size of the opening to eliminate options, and never proceeds to the actual decisive factor: precisely simulating Jay's first-person perspective to determine which elements are truly visible or occluded along the sightline to Tom.

**Possible reasons for failure:**
- Misinterpretation of schematic scale: The models are less intentionally trained on large spatial scale (building scale or above) reasoning data. They fail to infer that a gap spanning dozens of human-widths in the image is effectively a wide opening.
- Weak mental rotation and simulation: Accurately answering requires constructing a 3D mental model, placing a virtual camera at Jay's position and then tracing rays toward Tom. Such capabilities remain weak in frontier models.
- Failure of local-to-global perspective mapping: The task demands bidirectional reasoning: using the global perspective to predict local first-person visibility and vice versa. Most VLMs handle one direction poorly and completely fail when both directions are required simultaneously.

Figure 22: Failure Case 1 Analysis.

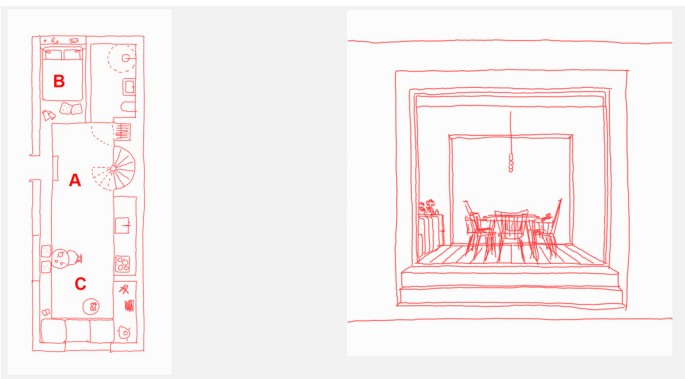

Which area is the perspective drawing framing?
A. Option A in the image          B. Option B in the image
C. Option C in the image          D. None of the labelled area ☑

**GPT-5 Prediction**

*We compare the floor plan (left) with the perspective drawing (right).*
*What the perspective shows:*
*A large opening (framed doorway or wall opening)*
*Steps leading up into a space*
*Inside that space:*
*A dining table with multiple chairs*
*A pendant light hanging from the ceiling*
*A sideboard / cabinet on the left*
*This is clearly a dining area.*
*Match with the plan:*
***Area A***
*Long corridor-like living space*
*Contains seating + spiral stair*
*Not a dining room → doesn't match*
***Area B***
*Bathroom area (sink + bathtub)*
*Definitely not shown in perspective → doesn't match*
***Area C***
*Contains a dining table with multiple chairs*
*A pendant light above the table*
*Adjacent to the large opening at the end of the unit*
*This matches the perspective exactly*
*Thus, the perspective drawing frames Area C*

**Analysis**

This is a failure of **fine-grained visual comparison** and **precise spatial layout verification**. The model made some coarse observations presented in both Area C and the perspective view. However, key differences in the perspective such as the number and location of chairs, wardrobes on the left, steps in front of the table are ignored by the model.

**Possible reasons for failure:**
**-** Loss of spatial precision: Models excel at global layout but rapidly lose fine positional and counting accuracy.
**-** No systematic verification loop: Humans subconsciously run a mental "spot-the-difference" process (count chairs, check what's behind table, look left for closet, notice steps, etc.). Models almost never iterate or explicitly check for absences; they accept the first plausible match.

Figure 23: Failure Case 2 Analysis.

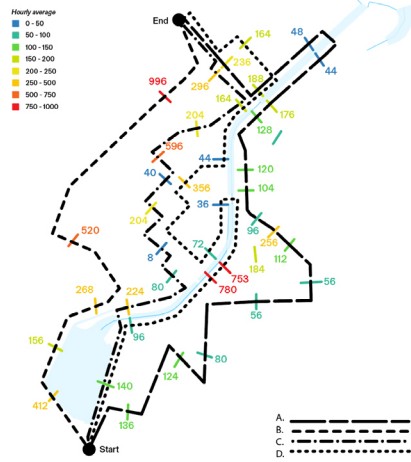

Above is a 'gate count' diagram, quantifying pedestrian flows through specified 'gates'. Which route from start to end should pedestrians take to meet as many people as possible?

A. Option A in the image
B. Option B in the image
C. Option C in the image
D. Option D in the image ☑

**GPT-5 Prediction**

*Why Route A has the highest encounters*
*You want the path from Start → End that passes through the largest total pedestrian counts, i.e., the route that intersects the warmest colors (orange/red = 500–1000 people per hour).*
*Key high-flow gates:*
*996 (dark red)*
*780, 753 (red)*
*596, 520 (orange)*
*Several 250–400 yellow gates*
*Which route intersects these?*
*Route A (long-dash) passes through:*
*780 (red)*
*753 (red)*
*596 (orange)*
*520 (orange)*
*296, 268, 256, 236, 204, 184, etc.*
*This route consistently stays on the highest-flow corridor, cutting through the warm-colored gates.*
***Other routes***
*B, C, D run through mostly greens and blues (low flows), intersecting only sparse mid-yellow segments.*
*They all avoid the key red/orange hotspot corridor.*
*Final Answer: A*
*Route A allows pedestrians to meet the maximum number of people.*

**Analysis**

The model's **reasoning is flawed**. It incorrectly assumes that route A maximises pedestrian encounters simply because it "cuts through the warm-colored gates," while routes "B, C, and D pass mostly through green and blue" gates. In reality, the correct approach is to sum the actual number of pedestrians at each gate along every route. When calculated properly, route D yields the highest total.

**Possible reason for failure:**
- Over-reliance on high-level semantic priors: Once the model notices color patterns, it activates a learned bias that warm colors correlate with crowds, a statistically valid prior from training data. This strong prior overrides the weaker, noisier signal from actually parsing the numbers.

Figure 24: Failure Case 3 Analysis.

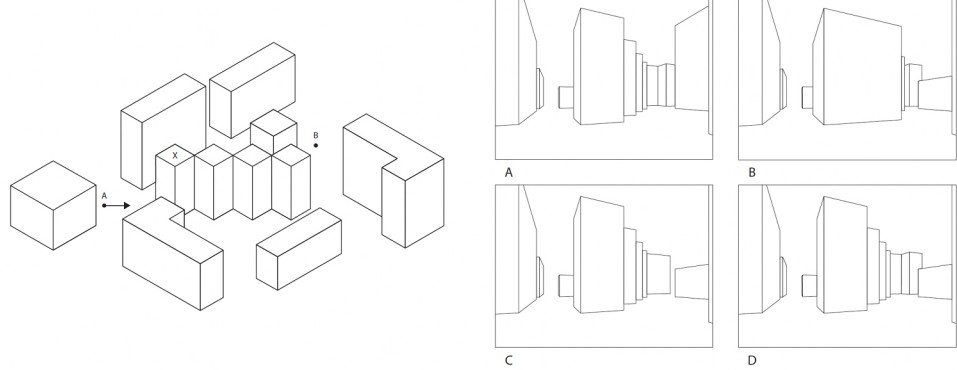

When standing at point A, looking in the direction where the arrow is pointing, which is the correct view?
A. Option A in the image          B. Option B in the image
C. Option C in the image          D. Option D in the image  ☑

**GPT-5 Prediction**

*You are standing at point A, looking toward point X (the small block at the front of a long, staggered corridor of blocks).*
*From A's position:*
*On the left, you see the large detached cube very close.*
*Immediately after that on the left, you see a tall rectangular block.*
*The corridor ahead contains a sequence of blocks, each slightly shifted, getting progressively smaller in apparent size due to perspective.*
*On the right side, a long L-shaped block begins close to you and continues inward.*
*The view is a deep corridor composed of many receding vertical elements.*
*Among the four given views:*
***Why A, B, and D are incorrect***
*A shows the rightmost block much too large and too close.*
*B shows the central corridor blocked by a huge block that does not match the plan.*
*D has the right-hand blocks appearing in the wrong sequence and proportion.*
***Why C is correct***
*Shows:*
*A large block on the left close by (matches the big cube near point A).*
*A tall block also on the left just behind it.*
*A long sequence of receding blocks straight ahead.*
*A right-hand block positioned at the correct depth.*
*Everything aligns with the 3D layout.*

**Analysis**

The model **fails to detect critical details**. The difference between options C and D is extremely subtle: the only distinction lies in the shape of the farthest building. From the bird's-eye view on the left, it is clearly an L-shaped building. Option C incorrectly renders it as a straight I-shape, whereas option D accurately depicts it as an L-shape.

**Possible reasons for failure:**
- Positional bias toward central/foreground objects: The models often prioritise foreground or center elements, effectively ignoring low-salience background details.
- Lack of explicit spatial reasoning training: The model may have "seen" the pixels but never learned to treat exact building match as decision-critical, so it defaulted to a superficial foreground match that looked plausible.

Figure 25: Failure Case 4 Analysis.

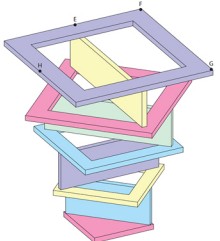 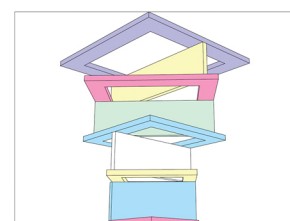

Consider the diagram on the left. Which color is the white wall in the structure in the image on the right according to it?

A. green    B. purple ☑    C. pink    D. blue

**GPT-5 Prediction**

*Reasoning*
*In the right-side perspective image, the white wall is a vertical wall located roughly in the middle of the tower stack.*
*To identify which colored wall it corresponds to in the left 3D diagram, we match:*
*- its height level*
*- its orientation*
*- its location relative to the colored frames*
*1. Identify the white wall in the right image*
*In the right image, the white wall stands between:*
*a blue frame below*
*a pink frame above*
*It is clearly the vertical wall at the third tier from the bottom.*
*2. Look at the left diagram*
*Find the third tier from the bottom:*
*Bottom: pink*
*2nd: blue*
*3rd: green ← the only vertical wall at that level.*
*It has the exact same orientation and width as the white wall in the right perspective.*
*Therefore, the white wall in the perspective corresponds to the green wall.*
*Correct answer: C. green*

**Analysis**

The model **hallucinates** the wrong neighboring colors for the white wall. The reasoning is sound, but it arrives at the wrong answer due to a clear perceptual error: it misidentifies the position of the white wall. In the actual image (right side), the white wall is positioned between a yellow frame below and a blue frame above.

**Possible reason for failure:**
- Local context confusion during patch-based processing: thin vertical features like walls can straddle patch boundaries or occupy only a few pixels within a patch, causing their adjacent colors to be incorrectly associated after tokenization and pooling.

Figure 26: Failure Case 5 Analysis.

