# OpenReview forum: "SpaCE-Eval: A Benchmark for Real-World Multi-Modal Reasoning"
_ICLR.cc/2026/Conference — ICLR 2026 Poster_

### Official Review · Reviewer_Mgud · 2025-10-26

**Soundness:** 2
**Presentation:** 2
**Contribution:** 2
**Rating:** 6
**Confidence:** 4

**Summary:**

SpaCE-Eval introduces a new visual question answering (VQA) benchmark designed to evaluate multi-modal large language models (MLLMs) on real-world spatial reasoning, commonsense knowledge, and environment interaction. The dataset features 1,139 high-quality, human-created diagram-question pairs spanning diverse spatial scales (from objects to urban spaces) and includes both textual and visual answer options. Extensive evaluation of leading MLLMs reveals significant gaps in spatial reasoning and simulation, especially as spatial scale increases, highlighting the need for more advanced real-world reasoning capabilities in AI models.

**Strengths:**

- Real-World Focus & Diversity: Covers a wide range of spatial scales and real-world scenarios, moving beyond object-level or synthetic tasks
- Rigorous Data Curation: All diagrams are newly created by human experts, eliminating data contamination and ensuring visual diversity.

**Weaknesses:**

- Need for artifacts: The paper identifies gap in MLLMs, provides a benchmark to help evaluate but should also provide basic artifacts like reasoning traces and training corpus that can be used to do SFT/RL and mitigate this gap, and further confirm the paper's hypothesis.
- Expand Task Formats: While broad in spatial scale, incorporating open-ended, multi-step, or interactive tasks (e.g., navigation, planning, or manipulation) to better reflect real-world reasoning demands would help the benchmark remain strong and relevant.
- Diagnostic Error Analysis: Provide more granular, qualitative analysis of failure cases, especially for abstract and large-scale spatial reasoning, to guide model and dataset improvements.

**Questions:**

Flags identified and covered above in weakness section

---

> ### Author Response · Authors · 2025-11-21
>
> Dear reviewer Mgud,
>
> Thank you for the thoughtful review and for recognising the benchmark’s real-world focus, diversity across spatial scales, and rigorous human-created data curation. Below, we address each of your comments.
>
> 1. Need for artefacts: We fully agree that releasing training material would be valuable for the community. However, we did not additionally release reasoning traces or a pre-collected training corpus (which would go beyond the scope of most ICLR benchmark papers), as SpaCE-Eval is intentionally designed as an evaluation-only benchmark. Training on the benchmark would compromise its purpose as a contamination-free, reasoning-intensive test set. That said, we understand the need for resources that advance the field. We are preparing a separate non-overlapping training dataset. We have added the discussion in the Section 5 Limitations and Future Work.
>
> 2. Expanding task format: We agree that navigation, planning, and manipulation tasks would further enrich real-world spatial reasoning coverage. Our rationale for starting with multiple-choice VQA is that it allows precise, objective evaluation without requiring subjective judgment or model-specific scoring, and avoids the inconsistencies of free-form language outputs, allowing systematic comparison across many MLLMs.  We also highlight that the **Environment Interaction** category is specifically designed to assess these dynamic, real-world demands. This category includes subcategories like *Mobility in Space (challenging models to select or plan navigation) and Design-Environment Interaction (asking questions from a decision-maker's view, e.g., how to set up a space to achieve desired goals)*. For example, the Environment Interaction task shown in Figure 1  requires multi-step planning in an anti-clockwise direction. The Mobility in Space example in Figure 16  requires flow analysis in an urban context. These already serve as proxies for the planning and reasoning required by autonomous agents.
>
> 3. Diagnostic Error Analysis: We completely agree that this would strengthen the paper. In the newly uploaded PDF, we have added typical failure cases with in-depth analysis in a new section: A.0.4 Detailed Failure Case Analysis. It demonstrates how the model failed to obtain the correct answer, and explains what might be the possible reasons for the failure to guide model and dataset improvements.

---

### Official Review · Reviewer_8eWj · 2025-10-30

**Soundness:** 3
**Presentation:** 3
**Contribution:** 3
**Rating:** 4
**Confidence:** 4

**Summary:**

This work proposes an expert-designed and peer-reviewed dataset aimed at evaluating the spatial reasoning abilities of various LLMs. This dataset comprises real-world VQA tasks across a range of contexts. Model outcomes show that for real-world spatial reasoning and use of LLMs in complex environment-scale spaces and creation of autonomous agents that can survive in the real world, these models require a great deal of enhancement and improvements to the reasoning process, and this work establishes this dataset as a human-verified benchmark to assess the same.

**Strengths:**

1. The dataset is well-curated, with a clear and comprehensive process for design, review, and modification.
2. The newly-designed nature of these problems eschews concerns about data leakage
3. Table 1 is a comprehensive comparison against similar datasets
4. The analysis section provides a useful look at model performance patterns and interpretations.

**Weaknesses:**

1. The data curation and review process are vital to the creation and success of this dataset. While the methodology is reasonable, there is no discussion of the soundness and reliability of this process - especially regarding the following;
- 1.1. The instructions provided for the design process, any rubrics that were followed, etc
- 1.2. Rubric for the review process, and how sounds this process was in terms of annotator agreement. This is especially relevant to the meta-annotators.
- 1.3 Examples of questions that did not pass quality control.

**Questions:**

Suggestions:
1. This work would benefit from demonstrating the question creation, review, and modification process for examples in the appendix, including questions that did not pass quality control
2. Fig 2 contains a typo in the diagram: SpeCE -> SpaCE
3. An exploration of how the quality control and review process was kept consistent and sound, and how annotator agreement was measured to ensure no bias or high variance in quality decisions, would lend more credibility to the curated dataset

Questions:
1. Given the small set of problem creators (51) all from the same background (university students), how do you ensure that problems are diverse not only in context and content, but also complexity?

---

> ### Author Response · Authors · 2025-11-21
>
> Dear reviewer 8eWj,
>
> Thank you for your constructive feedback. We address all your concerns below:
>
> **Diversity of the dataset**
>
> The diversity in complexity emerges naturally from the dataset design: multi-scale spatial reasoning, multi-step inference, twelve distinct subcategories, visual-only options, etc, each introduces fundamentally different levels of complexity. This diversity is also reflected empirically in the varied evaluation results of the same model across different spatial scales, subcategories, and question formats (visual-only or not) in Table 2 and Figure 4.
>
> Although the 51 contributors are all university students, they are not homogeneous. Instead, they represent multiple nationalities, cultural backgrounds and design traditions. We intentionally chose students with design (mostly architecture-related) backgrounds because they are trained to possess strong spatial abilities, including the ability to create high-quality visual representations of the physical world from scratch.  Before diagram creation, they studied a broad range of existing diagrams of various built environments in different countries across the planet and were trained to formulate questions centred on the subcategories. Each pair of contributors worked independently, bringing their own studio experiences and visual styles. In addition, we incorporated feedback from external reviewers outside the design disciplines to further broaden the diversity of perspectives represented in the final dataset.
>
> In the newly uploaded PDF, we have incorporated this content in section 3.2 Dataset Construction/Data curation
>
> **Quality control**
>
> To ensure that the quality control and review process was consistent and sound, we designed a multi-stage pipeline with shared rubrics and repeated alignment between contributors and meta-annotators. *We have elaborated Section 3.2 Dataset Construction/Data quality control in the manuscript*. We repeat the requirements and pipeline here:
>
> 1. All student contributors were given a written rubric that specified: (i) the diagrams must accurately represent information aligned with specific categories; (ii) the questions should be closely related to the diagram and the categories; (iii) reasoning process must be involved to answer the questions to avoid simple pattern match; (iv) linguistic or positional shortcuts should be avoided. (Section 3.2 Dataset Construction/Data curation)
> 2. During the data creation phase, contributors met weekly with the meta-annotators to review a subset of their diagram–question pairs. In these meetings, ambiguous cases were discussed in detail, and concrete examples were illustrated. This iterative feedback loop ensured that contributors gradually converged to a shared understanding of the rubric, leading to de facto annotator agreement over time rather than idiosyncratic interpretations by different students.
> 3. Volunteers from various backgrounds representing the general population are invited to review all image-question pairs and point out clarity issues, logic flaws, and any other errors for the contributors to refine the diagrams and texts accordingly.
> 4. A few dedicated reviewing sessions were conducted with external reviewers who were not involved in the initial data creation. Their independent perspective helped to surface hidden biases or systematic issues that might not be obvious to the original creators, further increasing the soundness and robustness of the dataset.
> 5. Multiple rounds of verification are conducted by meta-annotators. At this stage, 50 questions are adversarially rewritten to eliminate linguistic shortcuts where applicable. For example, options with features such as special length and sentiment that make them appear more likely to be the correct answer are modified. This involves a small proportion of questions. Each data entry is examined to ensure that referring to the visual input is necessary to answer the questions. During this process, 1468 questions remain unchanged.
> 6. Finally, all the data is filtered again by the authors, where the key selection criteria include clarity, accuracy, relevance, and diversity of the image-question pairs. In this process, 41 diagrams and 345 questions are excluded. As a result, 701 diagrams and 1139 questions that meet all requirements are kept.
>
> Together, the shared rubric, weekly alignment meetings, volunteer review, external review sessions, meta-annotator verification, and author filtering form a consistent and well-aligned quality control pipeline, aimed at minimising variance in quality decisions and ensuring that the curated dataset is both reliable and methodologically sound.
>
> **Examples not passing quality control**
>
> We’ve added some negative examples that did not pass the quality control and explanations in the Appendix (A.0.3) in the newly uploaded PDF.
>
> **Typo**
>
> Thank you for pointing out the typo. It has been corrected in the new PDF.

---

> > ### Author Response · Authors · 2025-12-01
> >
> > Dear reviewer,
> >
> > To further emphasise the robust data quality control pipeline, we added a new figure (*Figure 3: The pipeline of data quality control.*) in the manuscript in the newly uploaded PDF file. Together with the sections added and revised previously, we think we have addressed your concerns and significantly strengthened the paper.

---

### Official Review · Reviewer_xj7A · 2025-10-31

**Soundness:** 3
**Presentation:** 4
**Contribution:** 3
**Rating:** 6
**Confidence:** 4

**Summary:**

The paper introduces SpaCE-Eval, a new visual-question-answering benchmark designed to evaluate the real-world reasoning abilities of Multi-modal Large Language Models (MLLMs). The dataset consists of novel, human-drawn diagrams to prevent data contamination and is structured into three categories: Spatial Reasoning, Commonsense Knowledge, and Environment Interaction. Evaluations across various MLLMs reveal that the benchmark is highly challenging, especially in the spatial reasoning domain where models perform poorly. The results highlight a significant gap between current model capabilities and human-level spatial understanding, underscoring the need for improved reasoning in complex physical environments.

**Strengths:**

1.  **Strong and Clear Motivation:** The paper does an excellent job of identifying a critical and timely gap in existing MLLM evaluation. It correctly argues that reasoning in complex, environment-scale spaces is a fundamental prerequisite for real-world applications like robotics and autonomous agents, and that current benchmarks often fall short in this area.


2.  **Proactive Mitigation of Data Contamination:** The core methodological decision to create the entire dataset from brand-new, human-drawn diagrams is a significant strength. This approach directly addresses the pervasive problem of data contamination, where models may have already seen test images during pre-training, ensuring a more honest evaluation of their reasoning abilities.


3.  **Comprehensive and Well-Defined Task Taxonomy:** The benchmark is thoughtfully structured into three distinct and meaningful categories: **Spatial Reasoning (SR)**, **Commonsense Knowledge (CK)**, and **Environment Interaction (EI)**. Breaking these down further into twelve subcategories provides a fine-grained framework for analyzing model capabilities and pinpointing specific areas of weakness.


4.  **Novel Emphasis on Multi-Scale Spatial Reasoning:** A key contribution is the benchmark's explicit focus on reasoning across multiple spatial scales—from objects and rooms to buildings and urban contexts. This is a crucial aspect of real-world intelligence that is often overlooked in object-centric datasets, and SpaCE-Eval makes it a central part of its evaluation.


5.  **Focus on Abstract and Schematic Interpretation:** While the use of diagrams limits its "real-world" photographic realism, it is a strength in its own right. The ability to interpret abstract representations like floor plans, maps, assembly instructions, and infographics is a vital cognitive skill. This benchmark is specifically designed to test this form of abstract visual reasoning.

**Weaknesses:**

1.  **Superficial Analysis of Model Failures:** The paper correctly identifies that models fail at spatial simulation and abstract reasoning. However, the analysis in Section 4.3 is brief. It doesn't deeply investigate *why* these failures occur. Is it a limitation of the vision encoder, the language model's reasoning capabilities, the vision-language alignment, or something else?


2.  **Dataset Balance and Composition:** Figure 2 shows the dataset composition. "Building Space" accounts for nearly 50% of the data by scale. This heavy imbalance could mean the overall performance scores are disproportionately influenced by model capabilities on this specific scale, potentially masking poorer performance at other scales like urban or object.


3.  **In-Depth Bias Analysis:** Beyond a brief mention of linguistic shortcuts, the paper lacks a thorough analysis of potential biases in the dataset. This could include:
    *   **Answer Distribution Bias:** Is the correct answer (A, B, C, D) uniformly distributed? The paper claims it is, but doesn't provide the final distribution.
    *   **Content Bias:** Are certain objects, architectural styles, or cultural contexts overrepresented?
    *   **Negative Examples ("Probing"):** Including questions where the answer is "cannot be determined from the image" to test if models are hallucinating or guessing.

**Questions:**

1.  **Clarity of Category Definitions:** What are the precise, mutually exclusive definitions for the subcategories? For instance, how do you differentiate a task requiring "Spatial Interpretation" (reasoning about perspectives) from one requiring "Space Association" (associating spaces across different views)?


2.  **Meaning of CLIP Similarity Score:** The paper reports a mean CLIP similarity of 0.654 to argue for the novelty of the diagrams. How should this value be interpreted? Without a baseline (e.g., similarity scores within other VQA datasets or between random web images), it is difficult to judge whether 0.654 indicates true novelty or moderate similarity.


3.  **Reliability of the Automated Judge:** The paper uses GPT-4o-mini to verify model predictions that are not exact string matches. What was the measured accuracy of this automated judge? How was it validated to ensure it does not have its own biases or make systematic errors in judgment, which could affect the reported accuracy of the models being tested?

---

> ### Author Response · Authors · 2025-11-21
>
> Dear reviewer xj7A,
>
> Thank you for the detailed and enthusiastic review. We appreciate your positive assessment of the benchmark’s motivation, rigour, task taxonomy, and multi-scale design. Below, we address your concerns and clarify the requested points.
>
> 1. Analysis of Model Failures
>
> We agree that further diagnostic analysis is valuable. In the newly uploaded PDF, we have added a new session (A.0.4 Detailed Failure Case Analysis). It details how models arrive at incorrect answers and discusses the likely underlying causes of these failures to guide model and dataset improvements.
>
> 2. Dataset Balance and Composition
>
> You are correct that building-scale examples occupy a larger portion of the dataset. This reflects the natural prevalence of building-scale reasoning in daily environments. Despite this, each subcategory draws from multiple spatial scales. We have provided per-scale results in Figure 4 (c), and our analyses have treated each scale independently, ensuring that building-heavy distribution does not obscure scale-specific weaknesses. We will add an explicit note recommending that future users interpret overall scores together with per-scale metrics.
>
> 3. In-Depth Bias Analysis
>
> 3.1 The distribution of the correct answers being A, B, C, and D is 25.46%, 25.37%, 25.46%, and 23.71%, respectively. We have included the specific numbers in the updated manuscript.
>
> 3.2 Our benchmark intentionally focuses on the built environment, e.g. architectural spaces, urban contexts, and the everyday activities and cultures within them, because this domain constitutes a substantial portion of real-world human experience. This is not a narrow or culturally specific bias. Most human actions and spatial interactions occur in or around built structures, making this scope both representative and practically important. Within this domain, we also ensured diversity in spatial contexts, architectural forms, diagram styles, and environmental scenarios, drawing from diagrams and spatial representations across many regions and traditions. This focus, therefore, yields a dataset that is broad and globally relevant rather than stylistically or culturally skewed.
>
> 3.3 SpaCE-Eval already contains several such items. For example:
> - Figure 23 in the manuscript: option D: None of the labelled area.
>
> - t25_final_016: option A: none of the spaces.
> - t07_final_001: option C: cannot tell from the diagram.
> - t25_final_026: option B: The diagram provides insufficient information to determine.
>
> We agree that expanding this type would further strengthen the benchmark, and we will increase the number of such questions in the future release.
>
> 4. Clarity of Category Definitions
> Section 3.1 of the paper already provides formal definitions for all twelve subcategories, but we agree that some pairs, particularly Spatial Interpretation and Space Association, benefit from further clarification. In brief:
>
> - Spatial Interpretation focuses on understanding a single view by reasoning about perspective, orientation, depth, and visibility.
>
> - Space Association, by contrast, requires linking multiple distinct views (e.g., associating a floor plan with a sectional diagram, or matching a schematic drawing to a 3D perspective).
>
> We have revised Section 3.1 to make these distinctions more explicit. There are also examples of each subcategory in A.0.1 Dataset Category Details.
>
> 5. Meaning of CLIP Similarity Score
> We use CLIP similarity score to support the originality of our images, i.e. they are produced brand new, in contrast to collecting existing images online. In general, two images are considered “similar” when their similarity score is above 0.85-0.9. To provide a baseline, we downloaded 100 random images online and used the same method to calculate the similarity scores. The table below compares the results:
>
> | Data               | Mean  | 25th | Median | 75th |
> |--------------------|-------|------------------|--------|-----------------|
> | **Ours**           | 0.654 | 0.593            | 0.665  | 0.723           |
> | **Random Web Images** | 0.742 | 0.654            | 0.764  | 0.820           |
>
> 6.	Reliability of the Automated Judge
>
> Automated judging is commonly used in recent multimodal benchmarks. For example, HallusionBench (Guan et al., 2024) and MM-Vet (Yu et al., 2024) rely on GPT-4–assisted evaluation. In our evaluation, GPT-4o-mini is used to compare if the prediction matches the ground truth semantically, rather than purely open-minded subjective judgment, only when the models do not directly answer A, B, C or D.  To verify the reliability of our setup, we manually inspected 200 randomly sampled judgments produced by GPT-4o-mini and found only one incorrect case. This high level of agreement gives us confidence in the accuracy and consistency of GPT-4o-mini–assisted evaluation.

---

### Official Review · Reviewer_NpK9 · 2025-11-01

**Soundness:** 3
**Presentation:** 3
**Contribution:** 3
**Rating:** 6
**Confidence:** 3

**Summary:**

This paper introduces SpaCE-Eval, a new visual question answering (VQA) benchmark designed to evaluate the reasoning capabilities of Multi-modal Large Language Models (MLLMs) in real-world environments. The authors argue that existing benchmarks often fall short by focusing on object-scale understanding and static scenes, neglecting the complex, multi-scale reasoning required for practical applications. SpaCE-Eval addresses this gap with three core categories: Spatial Reasoning (SR), Commonsense Knowledge (CK), and Environment Interaction (EI). The benchmark consists of 1,139 high-quality image-question pairs based on brand-new, human-created diagrams to prevent data contamination. A comprehensive evaluation of numerousSOTA MLLMs on SpaCE-Eval reveals significant limitations, particularly in spatial reasoning, where even the best models perform poorly. The analysis highlights models' difficulties with larger spatial scales, tasks requiring spatial simulation, and reasoning with visual-only options, underscoring the benchmark's value in guiding future MLLM development.

**Strengths:**

1. The paper's greatest strength is the meticulous construction of the SpaCE-Eval benchmark. By commissioning brand-new diagrams from individuals with design backgrounds, the authors successfully avoid data contamination and create visually diverse and conceptually rich problems. The four-stage quality control process ensures the reliability and difficulty of the questions.
2. SpaCE-Eval is uniquely comprehensive. The three categories which are Spatial Reasoning, Commonsense Knowledge, and Environment Interaction, cover a wide and crucial set of abilities for real-world intelligence. The benchmark's focus on multiple spatial scales (from objects to urban environments) is a significant step beyond existing datasets.
3. The benchmark is demonstrably difficult for even the most advanced MLLMs, with the best model scoring only 56.37%. The paper provides deep insights into specific model failures, such as the stark performance gap between questions with textual vs. visual options, the degradation of performance on larger spatial scales, and the inability to perform mental spatial simulations. These findings are crucial for understanding the current limitations of MLLMs.

**Weaknesses:**

1. There is a discrepancy between the highest Spatial Reasoning score reported in the abstract (31.75% for claude-sonnet-4) and the one in Table 2 (42.25% for GPT-5). This should be clarified and corrected for consistency.
2. While the quality-over-quantity approach is commendable, the total dataset size of 1,139 examples is relatively small for an evaluation benchmark. When broken down into the twelve subcategories, the number of questions per subcategory can become quite small (e.g., Space Association is ~3.25%, which is only about 37 questions), which may limit the statistical significance of the fine-grained analysis.

**Questions:**

1. Could you please clarify the inconsistency regarding the top performance in the Spatial Reasoning category? Which value is correct: 31.75% as stated in the abstract or 42.25% as presented in Table 2?
2. Given the modest size of the dataset, do you have plans to expand SpaCE-Eval in the future? A larger number of examples would strengthen the statistical power of the results, especially for the more niche subcategories.
3. The analysis shows a significant performance drop for questions with purely visual options (26.58% accuracy). Is this poor performance uniform across the three main categories (SR, CK, EI), or is it particularly higher in one area, such as Spatial Reasoning, which seems intuitively more visual?

---

> ### Author Response · Authors · 2025-11-21
>
> Dear reviewer NpK9,
>
> Thank you for the thoughtful and encouraging review. We appreciate your recognition of the benchmark’s design rigour and novelty. We address your questions and concerns below.
>
> 1. Thank you for pointing out the inconsistency. We have rechecked the whole manuscript. Table 2 and Section 4.2 correctly report the best results as 56.37% overall and 42.25% in Spatial Reasoning, both achieved by GPT-5. We have uploaded an updated manuscript.
>
> 2. We agree that expanding SpaCE-Eval would further strengthen fine-grained statistical analyses, especially within smaller subcategories. In this initial release, all diagrams and questions were hand-crafted from scratch, so we prioritised quality over quantity. In fact, we are currently experimenting with a more efficient data curation pipeline and planning a more comprehensive version of the dataset with substantially more data entries. In the manuscript,  we have added Section 5 Limitation And Future Work to address this point.
>
> 3. We have updated Figure 4 to incorporate the per-category results in the new PDF.  The performance drop is uniform across all three main categories. We repeat the results here:
>
> | Model              | Category | Text Option | Visual Option | Mean   |
> |--------------------|----------|-------------|----------------|--------|
> | **gpt-4o**         | Overall  | 55.11       | 23.42          | 39.27  |
> |                    | SR       | 39.37       | 20.51          | 29.94  |
> |                    | CK       | 59.94       | 24.36          | 42.15  |
> |                    | EI       | 56.97       | 31.18          | 44.08  |
> | **gemini-2.5-flash** | Overall | 61.44       | 25.68          | 43.56  |
> |                    | SR       | 44.09       | 21.25          | 32.67  |
> |                    | CK       | 64.98       | 28.21          | 46.59  |
> |                    | EI       | 65.74       | 36.56          | 51.15  |
> | **Llama-4-Maverick** | Overall | 58.27       | 26.58          | 42.43  |
> |                    | SR       | 40.16       | 21.98          | 31.07  |
> |                    | CK       | 60.88       | 32.05          | 46.47  |
> |                    | EI       | 64.14       | 35.48          | 49.81  |

---

### Author Response · Authors · 2025-12-03

We are grateful to the reviewers for their insightful comments, which have allowed us to substantially strengthen the paper. Our revisions focus on enhancing the benchmark's rigour and analysis:

1. **Methodological Clarity:** We have added a more detailed description of the data quality control pipeline (Section 3.2) and extended to A.0.3 negative data samples.
2. **In-Depth Analysis:** We have performed a major expansion of our evaluation in Section A.0.4, providing an in-depth diagnostic analysis of model failure modes to guide future development of data and models.
3. **Enhanced Statistics:** We now include more fine-grained statistics and evaluation results across subcategories and spatial scales, along with revisions to clarify confusions regarding category definitions and data distribution.

These updates address all reviewers' concerns and solidify SpaCE-Eval as a robust and essential benchmark for real-world multi-modal reasoning.

---

### Meta-Review · Area_Chair_DjP1 · 2026-01-06

**Summary:**

This paper introduces SpaCE-Eval, a new visual-question-answering benchmark designed to evaluate Multi-modal Large Language Models (MLLMs) on real-world reasoning across spatial scales, commonsense knowledge, and environment interaction. The benchmark consists of 1,139 human-created diagram-question pairs to avoid data contamination, with rigorous multi-stage quality control. Experiments on a wide range of proprietary and open-source MLLMs reveal significant performance gaps, especially in spatial reasoning (best model: 42.25%) and on purely visual options, highlighting the limitations of current models in real-world spatial simulation and abstract reasoning. The paper makes a clear contribution by addressing a critical gap in existing benchmarks, which often focus on object-level understanding rather than multi-scale environmental reasoning.

I recommend accepting this paper. The primary strength of SpaCE-Eval lies in its methodological rigor regarding data contamination, a pervasive issue in the evaluation of large-scale MLLMs. By manually creating fresh diagrams, the authors provide a reliable testbed for "true" reasoning capabilities.

While the dataset size is modest, the trade-off for high-quality, hand-crafted, contamination-free data is acceptable for a benchmark paper. The authors were highly responsive during the rebuttal phase:

1. They significantly expanded the description of the Quality Control pipeline (addressing Reviewer 8eWj), detailing the shared rubrics, weekly alignment meetings, and adversarial rewriting.

2. They added a detailed Failure Case Analysis (Section A.0.4) and negative examples (addressing xj7A and Mgud), providing the requested qualitative depth.

3. They clarified that the "Building Space" dominance reflects real-world frequency and provided per-scale breakdowns to ensure fair evaluation.

The benchmark reveals a significant gap between current SOTA models and human performance in spatial reasoning, which will likely motivate further research in this domain. The concerns regarding size and lack of training data are noted as limitations, but do not overshadow the value of a clean, difficult evaluation set.

**Reviewer Concerns:**

The reviewers raised several valid concerns, primarily focusing on dataset size, methodology transparency, and depth of analysis:

1. Dataset Size and Balance: Multiple reviewers (NpK9, xj7A) noted that 1,139 examples is relatively small for a modern benchmark, potentially limiting statistical power when broken down into subcategories. Reviewer xj7A specifically pointed out an imbalance in spatial scales, with "Building Space" dominating nearly 50% of the data.

2. Quality Control Transparency: Reviewer 8eWj expressed concern regarding the lack of detail about the diagram creation rubric, annotator agreement, and the overall reliability of the curation process. They requested more transparency on how quality was maintained across diverse student contributors.

3. Failure Analysis: Reviewers xj7A and Mgud felt the initial manuscript lacked a deep diagnostic analysis of why models failed (e.g., vision encoder failure vs. reasoning failure), requesting more qualitative examples and "negative" data samples.

4. Artifacts: Reviewer Mgud suggested that providing a training corpus or reasoning traces would make the contribution more complete for improving models, rather than just evaluating them.

**Reviewer Scores:**

Three reviewers scored the paper 6 (marginally above acceptance threshold), and one scored it 4 (marginally below). All reviewers acknowledged the benchmark’s novelty, rigorous curation, and relevance to real-world reasoning. Confidence scores were generally high (3–4), indicating reviewers felt fairly to very confident in their assessments. The authors’ revisions, especially the added failure analysis and quality control details, addressed most concerns and strengthened the pape

---

### Decision · Program_Chairs · 2026-01-26

Accept (Poster)